# Efficient differentiation of human primordial germ cells through geometric control reveals a key role for Nodal signaling

**Kyoung Jo[1], Seth Teague[2†], Bohan Chen[1†], Hina Aftab Khan[1†], Emily Freeburne[1], Hunter Li[1], Bolin Li[1], Ran Ran[1], Jason R Spence[1,2,3,4], Idse Heemskerk[1,2,3,5]***

[1]Department of Cell and Developmental Biology, University of Michigan Medical School, Ann Arbor, United States; [2]Department of Biomedical Engineering, University of Michigan, Ann Arbor, United States; [3]Center for Organogenesis, University of Michigan Medical School, Ann Arbor, United States; [4]Department of Internal Medicine, Gastroenterology, University of Michigan Medical School, Ann Arbor, United States; [5]Department of Physics, University of Michigan, Ann Arbor, United States

**Abstract** Human primordial germ cells (hPGCs) form around the time of implantation and are the precursors of eggs and sperm. Many aspects of hPGC specification remain poorly understood because of the inaccessibility of the early postimplantation human embryo for study. Here, we show that micropatterned human pluripotent stem cells (hPSCs) treated with BMP4 give rise to hPGC-like cells (hPGCLC) and use these as a quantitatively reproducible and simple in vitro model to interrogate this important developmental event. We characterize micropatterned hPSCs up to 96 hr and show that hPGCLC populations are stable and continue to mature. By perturbing signaling during hPGCLC differentiation, we identify a previously unappreciated role for Nodal signaling and find that the relative timing and duration of BMP and Nodal signaling are critical parameters controlling the number of hPGCLCs. We formulate a mathematical model for a network of cross-repressive fates driven by Nodal and BMP signaling, which predicts the measured fate patterns after signaling perturbations. Finally, we show that hPSC colony size dictates the efficiency of hPGCLC specification, which led us to dramatically improve the efficiency of hPGCLC differentiation.

*For correspondence:
iheemske@umich.edu

†These authors contributed equally to this work

Competing interest: The authors declare that no competing interests exist.

## Editor's evaluation

This manuscript describes a powerful tissue culture model to study the early embryonic development of human primordial germ cells (PGC), the precursors of eggs and sperm. The study dissects the signaling pathways that direct the formation of PGC and provides important clues as to the origins of these cells during development.

## Introduction

Formation of primordial germ cells (PGC) is the first step in specification of the germline, the unique lineage through which genetic material is passed on to the next generation and potentially the key to understanding totipotency. Germline defects underlie numerous human diseases, most notably infertility (*Chen et al., 2017a*). Understanding PGC specification is therefore critical for both the fundamental understanding of human development and for its practical implications in disease. Yet, human

**eLife digest** In humans and other animals, eggs and sperm are unique cells that pass on genetic material to the next generation. They originate from a small group of cells called primordial germ cells that form early in life in the developing embryo. Several different signal molecules including ones known as BMP4, Wnt, and Nodal, instruct certain cells in the embryo to become primordial germ cells.

The process by which primordial germ cells are made in humans is very different to how primordial germ cells are made in mice and other so-called model animals that are commonly used in research. This has made it more challenging to uncover the details of the process in humans. Fortunately, new methods have recently been created that mimic aspects of how human embryos develop using human stem cells in a laboratory dish, providing an opportunity to gain a deeper understanding of how human germ cells form.

Jo et al. used a technique called micropatterning to control the shape and size of groups of human stem cells growing in a laboratory dish. Treating these cells with a signal known as BMP4 gave rise to cells that resembled primordial germ cells. The team then used this system as a model to study how primordial germ cells form in humans. The experiments found that reducing Wnt signals in stem cells stopped primordial germ cells from forming in response to BMP4, confirming that Wnt signals made by the cells in response to BMP4 are essential. However, this block was overcome by providing the stem cells with another signal called Nodal. This suggests that the role of Wnt signaling in primordial germ cell formation is in part indirect by switching on Nodal in stem cells.

Defects in eggs and sperm may lead to infertility, therefore, the findings of Jo et al. have the potential to help researchers develop new fertility treatments that use eggs or sperm grown in a laboratory from the patients' own stem cells. Such research would benefit from first developing a better understanding of how to make primordial germ cells.

germline specification remains an elusive process. Until recently, mammalian PGC specification was predominantly studied in mice. However, significant interspecies differences in PGC specification have been documented, particularly between rodents and primates but possibly also within the primates (*Kojima et al., 2017*; *Kobayashi et al., 2017*; *Kobayashi and Surani, 2018*). Because there is limited access to pre-implantation human embryos (*Hancock et al., 2021*) and it is not acceptable to study post-implantation human embryos, nonhuman primates and human pluripotent stem cell (hPSC)-based models of PGC differentiation have played a key role in advancing our understanding of this process. In vitro differentiation of PGC-like cells (PGCLCs) from hSPCs has been essential in revealing key aspects of human primordial germ cell-like cell (hPGCLC) specification such as the transcription factor network involving SOX17, PRDM1, and TFAP2C (*Irie et al., 2015*; *Chen et al., 2019*; *Kojima et al., 2017*). However, like many directed differentiation processes, PGCLC differentiation is inconsistent from batch to batch and cell line to cell line (*Chen et al., 2019*). This makes it difficult to systematically and quantitatively determine how and where PGCLCs arise in cell culture models. Although major progress has been made, much about germ cell specification remains poorly understood. For example, it is unknown whether human PGCs derive from the primitive streak (PS) as in mouse and pig, or from the amnion like in cynomolgus monkeys (*Kobayashi et al., 2017*; *Lawson et al., 1999*; *Sasaki et al., 2016*). It also remains unclear what the precise cell signaling requirements are that separate PGC specification from amnion, on the one hand, and mesendoderm, on the other hand.

Here, we used micropatterned hPSCs as a quantitatively reproducible system that allowed systematic interrogation of hPGCLC specification at single-cell resolution. Micropatterning enables spatial restriction of cell-substrate adhesion to control colony size and shape. Micropatterned human embryonic stem cells treated with BMP4 for 42–48 hr are a model system of human gastrulation, generating all three germ layers in concentric rings surrounded by another ring of extraembryonic-like cells (*Warmflash et al., 2014*). The inner domain consists of ectodermal or pluripotent cells depending on the differentiation media (*Chhabra et al., 2019*). Surrounding the inner domain is a ring of cells expressing PS markers such as TBXT (BRA) and EOMES. The outer ring of cells on the colony edge was initially thought to be trophectoderm (TE)-like due to its expression of CDX2 in the absence of TBXT but was later found to have features of both amnion and TE (*Chhabra and Warmflash, 2021*; *Minn et al., 2020*).

A final ring of SOX17-positive cells, roughly positioned between the extraembryonic cells and primitive-streak-like cells, was originally thought to be endoderm. However, these SOX17+ cells do not express the definitive endoderm marker FOXA2 (*Martyn et al., 2019b*). Moreover, they are positioned near the colony edge where BMP signaling is high, which contrasts with studies demonstrating that endoderm differentiation is improved by BMP inhibition (*Loh et al., 2014*). Additionally, murine endoderm is thought to arise from the anterior streak where BMP is low (*Nowotschin et al., 2019*). Here, we further investigate the identity of each of the cell types and report that this puzzle is resolved by the finding that the SOX17+ cells juxtaposed with the extraembryonic tissue at 42 hr are not endoderm but PGCLCs, confirming what was also recently reported in *Minn et al., 2020*. Although SOX17+ uniquely marks endoderm in the mouse, it is well known to be expressed in primate PGCLCs and the location of the PGCLCs in our system is consistent with mouse development, where PGCs arise in posterior streak at the interface with the extraembryonic tissue in a BMP-dependent manner.

We developed improved quantitative analysis of immunofluorescence (IF) data at the single-cell level based on a 3D image analysis pipeline integrating deep-learning-based segmentation. This enabled accurate assessment of the molecular signatures, spatial distributions, and sizes of cell populations. We combined this with scRNA-seq to confirm PGCLC identity and further found evidence of amniotic ectoderm identity of the outer ring. By carrying out temporal analysis up to 96 hr, we found that PGCLC populations persist and mature during this time window.

After confirming PGCLC specification, we carried out pharmacological and genetic perturbations to provide insight into the underlying signaling involved in this process. Although a requirement for Nodal in mouse PGC differentiation was demonstrated (*Senft et al., 2018b*; *Senft et al., 2019*; *Mulas et al., 2017*), directed differentiation of human PGCLCs has focused on BMP and Wnt and the precise roles and interplay of these pathways remain unclear (*Hancock et al., 2021*; *Kobayashi et al., 2017*). We confirmed a requirement for continuous BMP signaling for the first two days of hPGCLC differentiation but found that Wnt signaling is only required in a short time window before 24 hr and provide evidence that the primary role for Wnt is to induce Nodal. We showed that Nodal is required for hPGCLC induction and that exogenous stimulation of the Nodal pathway can rescue PGCLC induction when Wnt is inhibited. We found that the timing and duration of Nodal are critical in deciding between amnion-like, PGCLC and PS-like fates. In addition, we found that FGF/ERK signaling is essential throughout differentiation.

Finally, we investigated how PGCLC differentiation depends on colony size and found that by optimizing colony size we can generate PGCLCs with efficiencies of ~50% using BMP4 treatment alone. When 12 hr of pre-differentiation to an incipient mesoderm-like state (iMeLC) is included, as is typical in directed PGCLC differentiation, this number increases to 70% compared to other studies reporting 20–30% efficiency (*Sebastiano et al., 2021*; *Sasaki et al., 2015*).

## Results

### PGCLCs form on the interface between extraembryonic and primitive streak-like cells

Upon treatment with BMP4, at least four distinct cell fates arise in concentric rings by 42 hr in micro-patterned hPSC colonies with 500–1000 μm diameter (*Warmflash et al., 2014*). Cells expressing SOX17 have repeatedly been identified as endoderm (*Warmflash et al., 2014*; *Martyn et al., 2019b*). However, we found that these cells do not express the definitive endoderm marker FOXA2 (*Figure 1A*). In primates, SOX17 does not only mark definitive endoderm but also PGCs, which moreover form close to the interface of the posterior epiblast and amnion, with their precise origin in human still a point of debate (*Saitou, 2021*; *Hancock et al., 2021*). This suggests these cells could be PGC-like cells (PGCLCs) instead. To test this idea, we used IF to visualize the marker genes TFAP2C, PRDM1, and NANOG, which in combination are known to uniquely mark PGCLCs (*Tyser et al., 2021*; *Yang et al., 2021*). This confirmed our hypothesis and showed the reproducible presence of PGCLCs (*Figure 1B*), positioned between ISL1+ extraembryonic cells and EOMES+/TBXT + PS-like cells (*Figure 1—figure supplement 1A–D*).

To quantify the relationship between these markers from the IF data, we developed a 3D image analysis pipeline based on machine learning to handle multiple overlapping cell layers and automatically determine if cells express or co-express specific markers (see Materials and methods). Segmentation

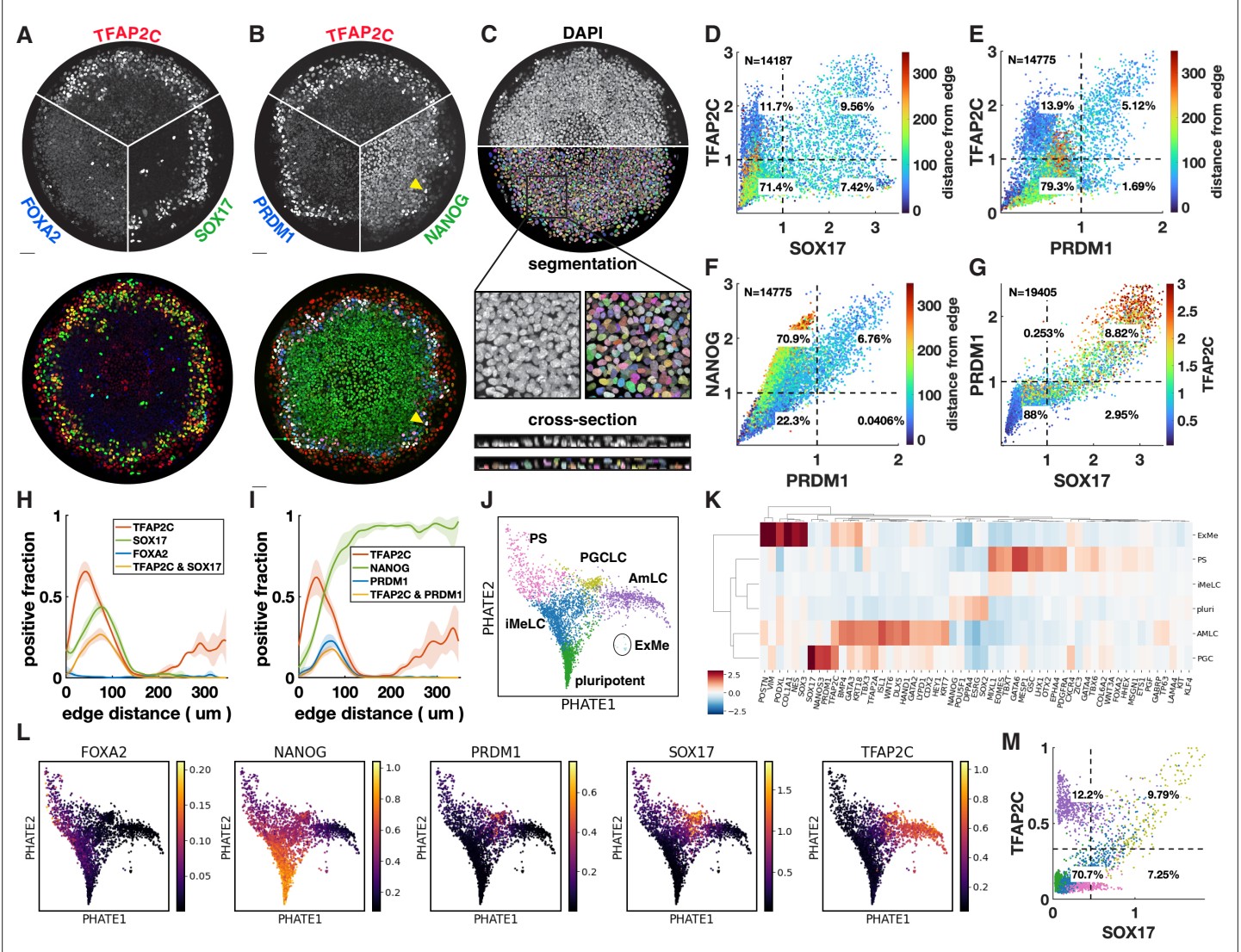

**Figure 1.** Primordial germ cell-like cells (PGCLCs) form at the interface between extraembryonic and primitive streak-like cells. (**A, B**) Immunofluorescence for different marker genes (maximal intensity projection along z). Yellow arrowhead in (**B**) points to higher NANOG expression in PGCLCs than pluripotent cells in the colony center. (**C**) Segmentation of nuclei based on DAPI staining. (**D–F**) Scatterplots of marker expression colored for radial position, normalized to threshold and log(1 + x) transformed, (**D**) corresponds to (**A**); (**E, F**) correspond to (**B**). (**G**) Scatterplot of PRDM1 vs. SOX17 colored for TFAP2C. (**H, I**) Spatial distribution of positive cells, dark lines represent the mean kernel density estimate of the positive fraction over four colonies, colored bands represent the standard deviation. (**J**) Clusters generated by Louvain. (**K**) Heatmap of differential expression between clusters (average z-scores) of genes associated with gastrulation. (**I**) PHATE visualization of scRNA-seq data showing denoised expression of markers used in (**B–I**) (raw in *Figure 1—figure supplement 2D*). (**M**) Scatterplot of TFAP2C vs. SOX17 from denoised scRNA-seq data (raw in *Figure 1—figure supplement 1P*), with colors matching clusters in (**J**). Scale bars 50 μm. All colonies are 700 μm diameter.

The online version of this article includes the following figure supplement(s) for figure 1:

**Figure supplement 1.** Quantitative relationships between marker genes with immunofluorescence (IF) and scRNA-seq.

**Figure supplement 2.** Additional scRNA-seq analysis, including correlation with CS7 human gastrula.

**Figure supplement 3.** Primordial germ cell-like cell (PGCLC) differentiation in different male and female cell lines.

(*Figure 1C*) allowed us to generate single-cell scatterplots of protein expression that show different co-expressing groups of cells in a manner similar to data from flow cytometry (*Figure 1D–G*).

Across multiple experiments, we found ~10–20% of cells to be SOX17+ with 50–60% of those also expressing TFAP2C (*Figure 1D*). About 5–10% of cells were found to be PRDM1+, which were mostly TFAP2C+, and all NANOG+ (*Figure 1E and F*). Moreover, PRDM1+ cells were nearly all SOX17+ (*Figure 1G*, *Figure 1—figure supplement 1E and F*), consistent with previous literature showing that

SOX17 is upstream of PRDM1 (*Kojima et al., 2017*). Most SOX17+ PRDM1+ cells expressed higher TFAP2C than SOX17+ PRDM1- (*Figure 1G*), and of SOX17+ TFAP2C+ cells, 80% were PRDM1+ (*Figure 1—figure supplement 1E–H*). Thus, PRDM1+ TFAP2C+ implies PRDM1+ TFAP2C+ SOX17+ NANOG+ and provides a conservative estimate of the PGCLC population while SOX17+ TFAP2C+ provides a similar but slightly higher estimate. Here, we will use both combinations to quantify the PGCLC population. Although average expression of NANOG in PGCLCs at 42 hr is similar to pluripotent cells, we observed that the highest levels of TFAP2C and NANOG occur in PGCLCs (*Figure 1D and F*), suggesting a positive feedback in the co-expression of these factors.

After identifying populations by thresholding markers, we visualized spatial patterning as the fraction of cells positive for a marker at some radius (*Figure 1H and I*). We found this to be a substantial improvement over average intensity profiles that have been used in studying micropatterned hPSCs (*Figure 1—figure supplement 1H and I*) because the relative magnitude of the markers in the graph becomes meaningful and eliminates the effect of background when positive cell populations are small (see FOXA2 in *Figure 1—figure supplement 1H*). Moreover, it allows simple visualization of the spatial profile of marker combinations like TFAP2C and SOX17 (*Figure 1H and I*).

We repeated quantitative analysis of IF for PGC markers with four different hPSC lines, both male and female (*Figure 1—figure supplement 3*). We found that all these form similar patterns although with some variability in the fraction of PGCLCs.

## scRNA-seq confirms PGCLC identity and shows extraembryonic cells resemble amnion

To further understand the identity of both the SOX17+ cells and other cells within the micropatterned hPSCs, we performed scRNA-seq and visualized our data using PHATE (*Moon et al., 2019*). This reproduced the known gene expression domains and organized them in a lineage tree-like layout with SOX2+ pluripotent cells at the bottom, a TBXT+ PS-like branch on the left, a ISL1+ branch on the right, with a group of SOX17+ cells between these two branches (*Figure 1—figure supplement 2A and B*). Diffusion components showed the SOX17+ cells more clearly as a third branch (*Figure 1—figure supplement 2C*).

To systematically evaluate gene expression in PGCLCs, we performed clustering using Louvain, which yielded six clusters (*Figure 1J*). We performed differential expression analysis to identify marker genes for each cluster, both for all genes and within a subset of marker genes relevant for gastrulation (*Supplementary files 1–3*). In addition, we found it instructive to visualize differential expression in a subset of marker genes that are commonly used to identify cell fate during gastrulation (*Figure 1K*). As expected, four of the clusters found by Louvain corresponded roughly to the cell groups identified using IF: pluripotent, PGCLC, extraembryonic, and PS-like. Confirming the IF, PGCLCs express NANOG and are marked by highly enriched expression of SOX17, PRDM1, TFAP2C, while FOXA2 expression was low and not in the same cells that expressed high SOX17 (*Figure 1K and L*). As expected, the PGCLC cluster also showed high expression of NANOS3, which is known to be uniquely expressed in PGCLCs (*Figure 1K*, *Figure 1—figure supplement 2E*).

The identity of the outer ring of cells has been a source of debate and is important in the context of PGCLC induction because PGCs have been found to derive from the amnion in cynomolgus monkeys, while their origin in human remains unclear (*Saitou, 2021*; *Hancock et al., 2021*). We argue that these cells are amnion-like and refer to them as AmLC.

The outer cells were previously found to express markers of both TE, including CDX2, GATA3, TP63, TBX3, and KRT7, but also genes associated with amnion such as TFAP2A, with little consensus on which genes specifically mark human amnion in vivo (*Chhabra and Warmflash, 2021*; *Minn et al., 2020*; *Chen et al., 2019*; *Sasaki et al., 2016*; *Knöfler et al., 2019*). Recently, ISL1 and BMP4 were suggested as key amniotic genes and GABRP and WNT6 as additional amnion markers (*Yang et al., 2021*). We found that the cells of the outer ring express all of the above (*Figure 1K*, *Figure 1—figure supplement 2E*), raising the question of whether this is a state between amnion and TE that is an artifact of the in vitro system. There is no published complete expression profile of both amnion and TE from a single human or even nonhuman primate embryo to validate the presumed markers and compare the two tissues. However, there is in vivo transcriptome data for human amnion from the CS7 human gastrula (*Tyser et al., 2021*), which contains an ectodermal cluster, including amniotic ectoderm marked by GABRP. We found that these cells express all the markers mentioned above

(*Figure 1—figure supplement 2F*). We also quantified the similarity between our clusters and those in the human gastrula dataset by cross-correlating gene expression (*Figure 1—figure supplement 2G*) and found strong correlation between our AmLC and the ectodermal cluster from *Tyser et al., 2021*. The placement of this cluster within the UMAP embedding, branching off between epiblast and PS close to the PGCs (Tyser Figure 1C), is also very similar to what is observed in *Figure 1J*. Furthermore, due to the way the sample was dissected it was unlikely to contain TE. This suggests that the outer ring on micropatterns is amniotic ectoderm, and that the expression of several markers that were thought to be TE is a feature of human amniotic ectoderm. However, until unambiguous in vivo data is published comparing amnion, TE and non-neural ectoderm, or functional data can be obtained, we cannot be completely certain of the identity of this cell population.

A fifth cluster was identified between the pluripotent cells and differentiated fate, expressing intermediate levels of both PS and pluripotency markers. This appears to be a transitionary state. In this context, we decided to name this intermediate state incipient mesoderm-like (iMeLC) as is used in directed differentiation of PGCLCs (*Sasaki et al., 2015*).

The sixth and final cluster of cells was very clearly distinct from other cells in the scRNA-seq data; however, very few cells were captured in this cluster, suggesting that these cells are rare within micropatterns. This cluster was enriched for ANXA1, POSTN, VIM, and PODXL (*Figure 1K*, *Supplementary files 1–3*), suggesting a yolk sac-mesoderm or extraembryonic mesenchyme identity. However, expected expression of GATA6 is missing. Moreover, it would be very surprising to find extraembryonic mesenchyme, which is thought to derive from the hypoblast, a tissue that is absent from our in vitro model. Correlation with CS7 gastrula data nevertheless did show significant correlation between this cluster and the YS mesoderm (*Figure 1—figure supplement 2G*), so this cluster is annotated as extraembryonic mesoderm (ExMe). Given the small number of cells for this cluster, expression data is very noisy and future investigation must confirm the consistent presence, identity, and origin of these cells.

## Immunofluorescence and scRNA-seq reveal similar quantitative gene relationships

We asked whether the gene expression relationships found using IF in *Figure 1D–G* could also be recovered when considering the same two genes in the scRNA-seq data, and whether the clusters obtained based on two markers correspond to the clusters identified in the full scRNA-seq dataset. While raw scRNA-seq data is too noisy to directly relate expression of two genes (*Figure 1—figure supplement 1L, O, and P*), after denoising using MAGIC (*van Dijk et al., 2018*), clear patterns emerged (*Figure 1M*). Thresholding using the same procedure used for IF produced nearly identical proportions of cells expressing SOX17 and/or TFAP2C with most of the cells in the SOX17+ TFAP2C+ quadrant belonging to the PGCLC cluster found by Louvain, demonstrating consistency between the data types and clustering procedures. *Figure 1M* further suggests that a significant fraction of SOX17+ TFAP2C- cells belong to the iMeLC cluster and are becoming PGCLCs.

It remains unclear whether human PGCs derive from amnion or posterior epiblast. Moreover, PGCs are thought to go through an incipient mesodermal state transiently expressing low levels of PS markers. We therefore also looked at co-expression of PGC markers with the PS markers EOMES and TBXT, and the amnion marker ISL1 (*Figure 1—figure supplement 1*).

EOMES is required to induce SOX17 during human PGCLC specification but is then rapidly downregulated (*Kojima et al., 2017*; *Chen et al., 2017b*). In the mouse, EOMES is not directly required for PGCLC induction, but it is required for EMT in gastrulation and specification of endoderm and cardiac mesoderm (*Senft et al., 2018a*; *Costello et al., 2011*; *Tosic et al., 2019*; *Arnold et al., 2008*). EOMES knockout hPSCs suggest that these functions in gastrulation are conserved in human (*Teo et al., 2011*; *Pfeiffer et al., 2018*). While in the overlay image it appears that ISL1 forms a shape boundary and has little overlap with EOMES and SOX17, the individual images and quantification show low EOMES and ISL1 expression in SOX17+ cells (*Figure 1—figure supplement 1A and J*). The scatterplot shows a striking inverse relationship between ISL1 and EOMES, suggesting mutual repression, with EOMES^low ISL1^low SOX17+ cells connecting EOMES+ ISL1 cells to EOMES-ISL+. This is consistent with a requirement for, but subsequent suppression of, EOMES in PGCLCs and places the gene expression profile of PGCLCs intermediate between amnion and PS. Similarly, we found that TFAP2C+ cells that co-express TBXT are mostly PRDM1+ and show a

strong correlation between TFAP2C and PRDM1 within this cluster (*Figure 1—figure supplement 1C and M*).

The relationships produced by plotting scRNA-seq data for the same genes look remarkably similar, particularly for ISL1 vs. EOMES (*Figure 1—figure supplement 1J and K*). Some differences do exist, for example, the gap between TFAP2C+ and TFAP2C- along the TFAP2C axis in the scRNA-seq data, which is not present in the IF data (*Figure 1M*, *Figure 1—figure supplement 1N*). There are several possible explanations for this: it may be a difference between RNA and protein levels, the gap could be an artifact of the denoising algorithm on the single-cell RNA-seq data, or the lack of a gap could be due to noise in the IF data. Nevertheless, these data show that consistent quantitative relationships can be recovered from these different types of data, which may inform mathematical models for the underlying gene regulatory networks (GRNs).

## PGCs are specified by 42 hr but continue to mature through 96 hr, while endoderm forms between 42 and 72 hr

We next asked whether the PGCLC population would persist and continue to develop past the 48 hr when BMP4-treated micropatterned colonies have so far been studied. In addition, we wanted to know whether definitive endoderm arises later in development. Therefore, we quantified the time course of TFAP2C, SOX17, and FOXA2 with 24 hr resolution up to 96 hr (*Figure 2*) and also included 42 hr in the analysis because this is the end point used in other experiments (*Figure 2D–F*). IF immediately revealed significant changes between 48 and 72 hr with a striking pattern of alternating clusters of PGCLCs (SOX17+ TFAP2C+) and endoderm (SOX17+ FOXA2+) appearing around the perimeter by 72 hr.

Our quantification showed that in contrast to 42 hr (*Figure 1A*) small numbers of FOXA2+ cells, both SOX17+ and SOX17-, emerge at 48 hr followed by a large increase in FOXA2+ SOX17+ cells between 48 and 72 hr, while FOXA2+ SOX17 were no longer present at 72 hr (*Figure 2C and D*). Quantitative analysis confirmed that FOXA2+ SOX17+ are TFAP2C- while FOXA2-SOX17+ are TFAP2C+, consistent with PGC and endodermal populations (*Figure 2C*). We conclude that endoderm is specified between 42 and 72 hr. In contrast, PGCs may be fully specified before 42 hr and do not appear to proliferate after that time since PGC numbers are stable between 42 and 72 hr. Between 72 and 96 hr, we observed no significant changes in either cell population.

Since the 72 and 96 hr time points have not been examined previously, we also looked at overall growth and morphology of the colonies over time. We found that the growth rate is gradually decreased from a 60% increase in cell number from 24 to 48 hr, to 38% from 48 to 72 hr and 5% from 72 to 96 hr (*Figure 2E*, *Figure 2—figure supplement 1A*). Because of the continued growth but stable PGC population, the percentage of PGCs goes down over time, which is reflected in the spatial distributions (*Figure 2—figure supplement 1B*). Looking at the 3D structure, we found that colonies become significantly thicker between 48 and 72 hr, forming either a multilayered structure or pseudostratified epithelium (*Figure 2—figure supplement 1C and D*). Peripheral endoderm and PGC clusters appear near the top of the colony while the scattered SOX17+ cells throughout the center of the colony are on the bottom of the colony. From 72 to 96 hr, the colony undergoes a slight morphological change, expanding outward beyond the borders of the micropattern and thinning in the colony center while the positioning of the endoderm and PGCs remains the same.

Finally, we asked whether PGCs mature over time. We measured the pluripotency markers NANOG and POU5F1 over time in pluripotent cells (PRDM1-NANOG+ POU5F1+) versus PGCLCs (PRDM1+ NANOG+ POU5F1+) and found that these markers are upregulated over time in PGCLCs to levels significantly higher than in pluripotent cells (*Figure 2G and H*, *Figure 2—figure supplement 2*) as has been previously observed (*Kojima et al., 2017*). As at 42 hr, the highest NANOG and POU5F1 levels at 48 hr are found in PRDM1+ cells even though the mean in PRDM1+ cells is not significantly higher than in pluripotent cells (*Figure 2—figure supplement 2C*). We also measured the expression of several more mature PGC markers using qPCR and found that DPPA3 (stella) and DDX4 (vasa) show an increasing trend between 48 and 96 hr, indicating that after their initial specification PGC development continues (*Figure 2F*, *Figure 2—figure supplement 1E*). Another mature PGC marker, DAZL, did not show significant expression, which is consistent with *Irie et al., 2015*, which detected DAZL only in embryonic gonadal PGCs. In this context, the increase in DDX4 is surprising since *Irie et al., 2015* also found DDX4 to only be expressed in gonadal PGCs and not in their hPGCLCs. We conclude

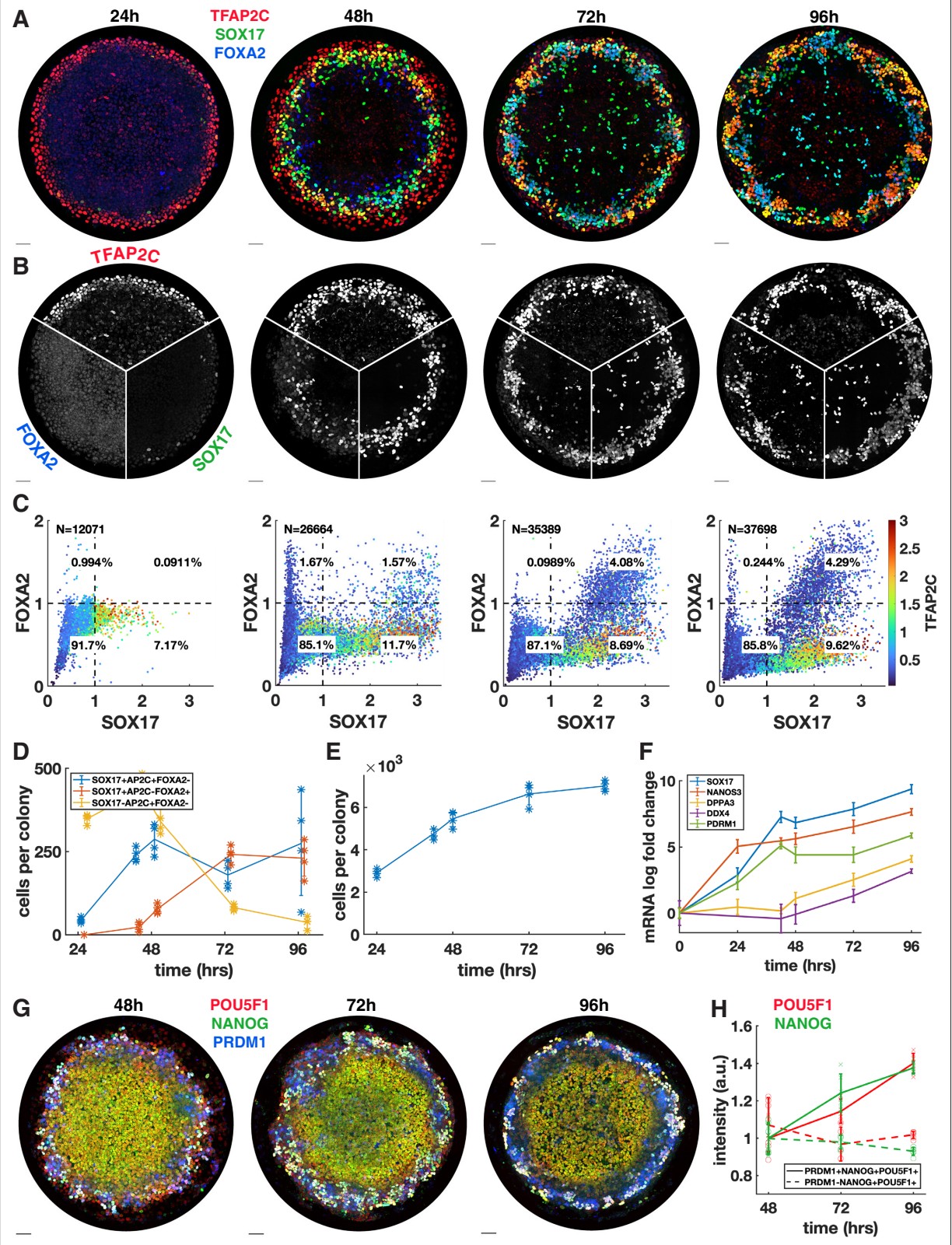

**Figure 2.** Primordial germ cells (PGCs) are specified by 42 hr but continue to mature while endoderm arises between 42 and 72 hr. (**A, B**) Immunofluorescence over time showing a stable PGC population and later emergence of endoderm. (**C**) Quantification of marker expression at different times showing the emergence of endoderm starting at 48 hr. (**D**) Absolute numbers of cell-expressing marker combinations corresponding to endoderm (red, SOX17+ AP2C-FOXA2+) primordial germ cell-like cells (PGCLCs) (blue, SOX17+ AP2C+ FOXA2-) and SOX17-AP2C+ FOXA2- (yellow). (**E**) Average

*Figure 2 continued on next page*

*Figure 2 continued*

cell number per colony over time. (**F**) qPCR data for PGC markers over time. (**G, H**) Immunofluorescence and quantification of pluripotency markers in PGCs over time. DAPI stainings corresponding to (**A, B**) are shown in *Figure 2—figure supplement 1*. Scale bar 50 μm.

The online version of this article includes the following figure supplement(s) for figure 2:

**Figure supplement 1.** Additional images and quantification for time series up to 96 hr.

**Figure supplement 2.** Additional images for pluripotency markers over time.

---

that PGCLCs in our system are stable and develop robust PGC-like gene expression over the course of 96 hr, similar to PGCLCs in other systems.

## PGCLCs share requirement for sustained BMP signaling with amnion-like cells

BMP and Wnt signaling are known to be important for PGC specification, and PGCLC specification is known to sensitively depend on the duration of exogenous Wnt activation, with prolonged Wnt activation leading to PS-like fates instead (*Kobayashi et al., 2017*; *Sasaki et al., 2015*). However, the specific timing of the interplay between these two pathways is not well understood. We asked whether BMP and Wnt act primarily through direct activation of PGC genes or indirectly through induction of secondary signals, and whether the timing and duration of these signals matters.

First, we inhibited BMP signaling after 24 hr with the BMP receptor inhibitor (BMPRi) LDN193189 (*Figure 3A–E*, *Figure 3—figure supplement 1A–F*). This led to a loss of PRDM1, a reduction in TFAP2C, and outward displacement of TBXT. Notably, it also gave rise to a new FOXA2+ SOX17- population at 42 hr (*Figure 3D and E*, *Figure 3—figure supplement 1E and F*), which, given the developmental stage and co-expression of TBXT, may be axial mesoderm.

Many of the effects of BMP4 are known to be indirect through the BMP-Wnt-Nodal cascade (*Chhabra et al., 2019*). To test whether the effect of BMPRi on PGC formation is direct or indirect, we next blocked Wnt production (Wnti) downstream of BMP by IWP2 at different times (*Figure 3F–J*, *Figure 3—figure supplement 1G*). Surprisingly, blocking Wnt production after 24 hr has almost no effect on PGC production, even though, as previously described, PS markers are significantly reduced (*Chhabra et al., 2019*). This suggests that endogenous Wnt signaling after 24 hr is involved in PS differentiation and that the effect of BMPRi on PGCLC specification after 24 hr is not due to WNT activation downstream of BMP but reflects a direct requirement for BMP. PGCLCs share this dependence on sustained BMP signaling with AmLC, which were previously shown to require continuous BMP signaling past 24 hr (*Chhabra et al., 2019*; *Nemashkalo et al., 2017*).

As expected, Wnt inhibition for the full duration of the experiment eliminates expression of TBXT and PRDM1. Strikingly, blocking Wnt signaling after 12 hr also led to loss complete loss of PRDM1, indicating that Wnt signaling between 12 and 24 hr after addition of BMP4 is critical (*Figure 3G–J*). We also noticed a reduction in the width of the TFAP2C expressing outer ring in these conditions (*Figure 3G and H*). However, quantification showed that the number of TFAP2C-positive cells was not reduced, implying that the cells were more densely packed (*Figure 3J*). This suggests that the change to a more spread morphology that is typically observed in the outer cells depends directly or indirectly on Wnt signaling.

## Nodal and FGF signaling are required on the second day of PGCLC specification

Directed differentiation protocols for PGCs typically consist of a brief period of exposure to PS-inducing signals Wnt and Nodal to induce an iMeLC followed by BMP. Therefore, we asked whether the other signals that are required to specify mesoderm: FGF and Nodal, are also only required during the first 24 hr to induce PGCs.

First, we inhibited FGF signaling with the FGF receptor inhibitor (FGFRi) PD173074 and MEK signaling (MEKi) with PD0325901 after 0 and 24 hr (*Figure 3K–O*, *Figure 3—figure supplement 1H and I*). The effects were very similar, suggesting that FGF specifies cell fate primarily through the MEK/ERK pathway and that the MEK/ERK pathway is primarily activated by FGF. When inhibiting FGF/ERK at 0 hr, TFAP2C expression becomes uniform throughout the colony and SOX17 expression is eliminated. When adding the inhibitor at 24 hr, TFAP2C expression expands inward and goes up in

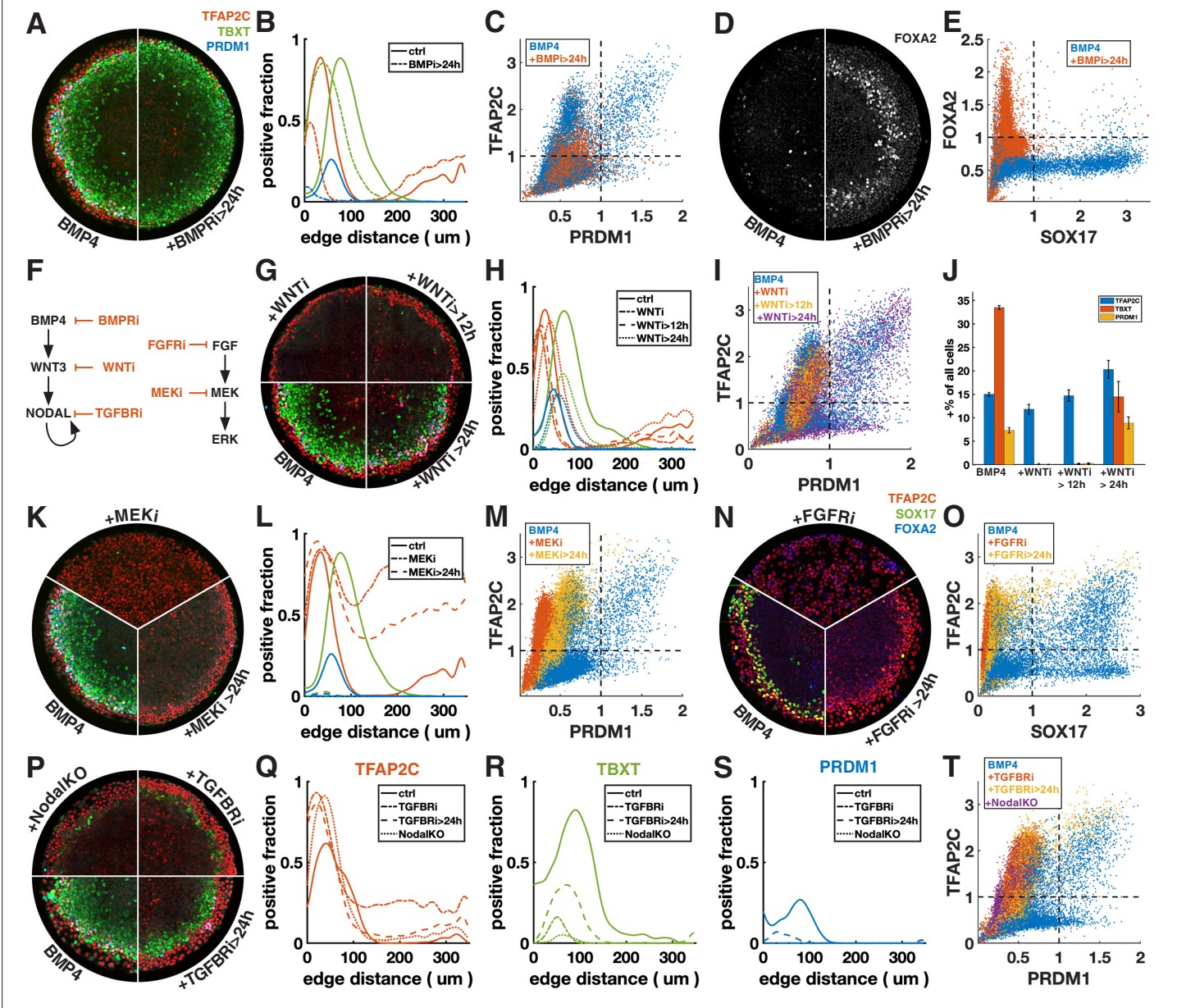

**Figure 3.** Primordial germ cell-like cells (PGCLCs) require sustained BMP, Nodal, and FGF but only brief Wnt signaling. Each row shows staining and quantification of PGC markers after perturbation of different pathways. Error bands in spatial distributions are omitted for clarity but are similar in magnitude to *Figure 1C and E*. (**A–E**) BMP4-treated colonies with or without BMP-receptor inhibition after 24 hr (BMPRi, LDN193189, 250 nM) shows loss of PGCLCs (**A–C**) and emergence of FOXA2+ SOX17- population. (**F**) Diagram of signaling hierarchy (black) and perturbations in this figure (red). (**G–J**) Wnt inhibition using IWP2 5 μM after 0, 12, and 24 hr showing PGCLC specification only requires Wnt signaling between 12 and 24 hr. (**K–O**) Inhibition of FGFR (PD-173074, 1 μM) or MEK (PD-0325901, 5 μM) at 0 and 24 hr showing complete loss of PGCLCs in both cases. (**P–T**) Inhibition of Nodal receptors (TGFBRi, SB-431542, 10 μM) at 0 and 24 hr and Nodal knockout (NodalKO) showing complete loss or severe reduction of PGCLCs. Images of each channel separately are shown in *Figure 2—figure supplement 2*. Scale bar 50 μm.

The online version of this article includes the following figure supplement(s) for figure 3:

**Figure supplement 1.** Additional images and quantification for signaling perturbations in *Figure 3*.

the colony center but is not uniform and a small number SOX17+ TFAP2C+ PGCLCs is present. Our observation of inward expansion of TFAP2C is consistent with findings in zebrafish, where TFAP2C was found to be a direct target of BMP whose expression is excluded from the margin by FGF/ERK signaling (*Rogers et al., 2020*), suggesting conserved regulation by signaling of this gene despite diverging functions.

Next, we inhibited Nodal signaling with the TGF-beta receptor inhibitor (TFGBRi) SB431542 (*Figure 3P–T*, *Figure 3—figure supplement 1J*). Nodal inhibition for the full duration of the experiment eliminates PRDM1 completely and largely eliminates TBXT, while TFAP2C goes up inside the colony, similar to FGF inhibition, suggesting that either FGF and Nodal both restrict BMP response to the edge, or that one of these signals modulates the other. Inhibition at 24 hr severely reduces PRDM1 and TBXT expression but leaves a clearly defined ring of TBXT expression. This indicates that while Wnt signaling is only required during the first 24 hr, both Nodal and FGF are also required at early and later times. Because there is TGF-beta in the differentiation media, TGFBRi not only blocks endogenous Nodal, but also the exogenous TGF-beta. To distinguish these effects, we examined Nodal-/- cells (*Figure 3P–T*; *Chhabra et al., 2019*). While the panel of markers appeared similar to TGFBRi-treated WT cells, TFAP2C expression in the center did not increase as much, indicating that in the absence of endogenous Nodal, low doses of TGFb in the differentiation media suppress TFAP2C and possibly BMP response more generally in the colony center.

## Exogenous activin rescues PGCLCs from endogenous Wnt inhibition or Nodal knockout in a dose- and time-dependent manner

Given that Wnt induces Nodal, it is possible that the main role of Wnt for PGCLC specification is to induce Nodal. While it is known that Nodal alone does not induce differentiation, it is unclear what happens in combination with BMP. We therefore tested whether we could rescue the loss of PGCLCs after Wnt inhibition by adding Activin to exogenously stimulate the Nodal pathway. PGCLC induction was indeed partially rescued by intermediate doses of Activin with robust expression of SOX17 and TFAP2C (*Figure 4A and B*). This is surprising considering the literature that has emphasized the role of Wnt signaling in hPGCLC induction (*Hancock et al., 2021*; *Kobayashi et al., 2017*). To support the idea that we are replacing endogenous Nodal downstream of Wnt, and not somehow restoring Wnt signaling downstream of Nodal through an unknown feedback loop, we stained for the Wnt target LEF1, which has been previously used as a readout of Wnt signaling in micropatterned colonies (*Martyn et al., 2019a*). This showed that Wnt signaling is not restored by Activin treatment (*Figure 4C and E*). We also compared Activin rescue of PGCLCs after Wnt inhibition at 0 and 12 hr and found that PGCs are rescued to a similar extent at 12 hr (*Figure 4C*).

We repeated the experiments with a different Wnt inhibitor, IWR-1, which stabilizes AXIN to inhibit canonical Wnt signaling, whereas IWP2 used in earlier experiments acts on PORCN to block Wnt secretion, thereby inhibiting both canonical and noncanonical Wnt signaling. A high dose of IWR-1 (50 μM) inhibited PGCLC differentiation, although some LEF1 remained (*Figure 4F*, *Figure 4—figure supplement 1C and D*). We again observed rescue of PGCLCs by Activin treatment without an increase in LEF1, exceeding the number of PGCLCs with IWP2 + Activin as well as with BMP4 only (*Figure 4F–H*). To make sure that under these conditions SOX17+ TFAP2C+ still is a good proxy for PGCLCs and implies the presence of TFAP2C+ PRDM1+ NANOG+ cells, we also stained for these markers and found that their expression is also restored, although at lower levels than expected for IWP2 (*Figure 4I and J*, *Figure 4—figure supplement 1E and F*).

The increased number of PGCLCs after Activin treatment with IWR-1 compared to IWP2 suggests either a different efficacy in inhibiting WNT signaling or a role for noncanonical WNT signaling, which is not inhibited by IWR-1. The inability of IWR-1 to inhibit PGCs at a lower dose (*Figure 4—figure supplement 1C*) and the LEF1 remaining after IWR-1 treatment (*Figure 4H*) suggested the former. To further test this hypothesis, we lowered the dose of IWP2 from 5 μM to 1 μM. This lower dose of IWP2 still completely inhibited PGC differentiation but was rescued to a much greater extent by Activin treatment (*Figure 4—figure supplement 1G*). Our results suggest that IWR-1 and IWP2 do not completely inhibit Wnt signaling even if in the absence of Activin they completely block differentiation to PGCLCs and PS-like fates, and that the remaining low levels of Wnt signaling correlate with PGCLC rescue by Activin treatment. The simplest interpretation of our results is that a low level of Wnt signaling is needed for PGCLC competence while a much higher level is needed indirectly to induce Nodal. Although different drugs and doses may each bring Wnt levels below those needed to induce Nodal, the remaining Wnt activity may control the size of the population that is competent to become PGCs when treated with Activin. Although the interplay between Wnt and Nodal is nuanced, we conclude that a significant part of the effect of Wnt is indirect due to its induction of Nodal.

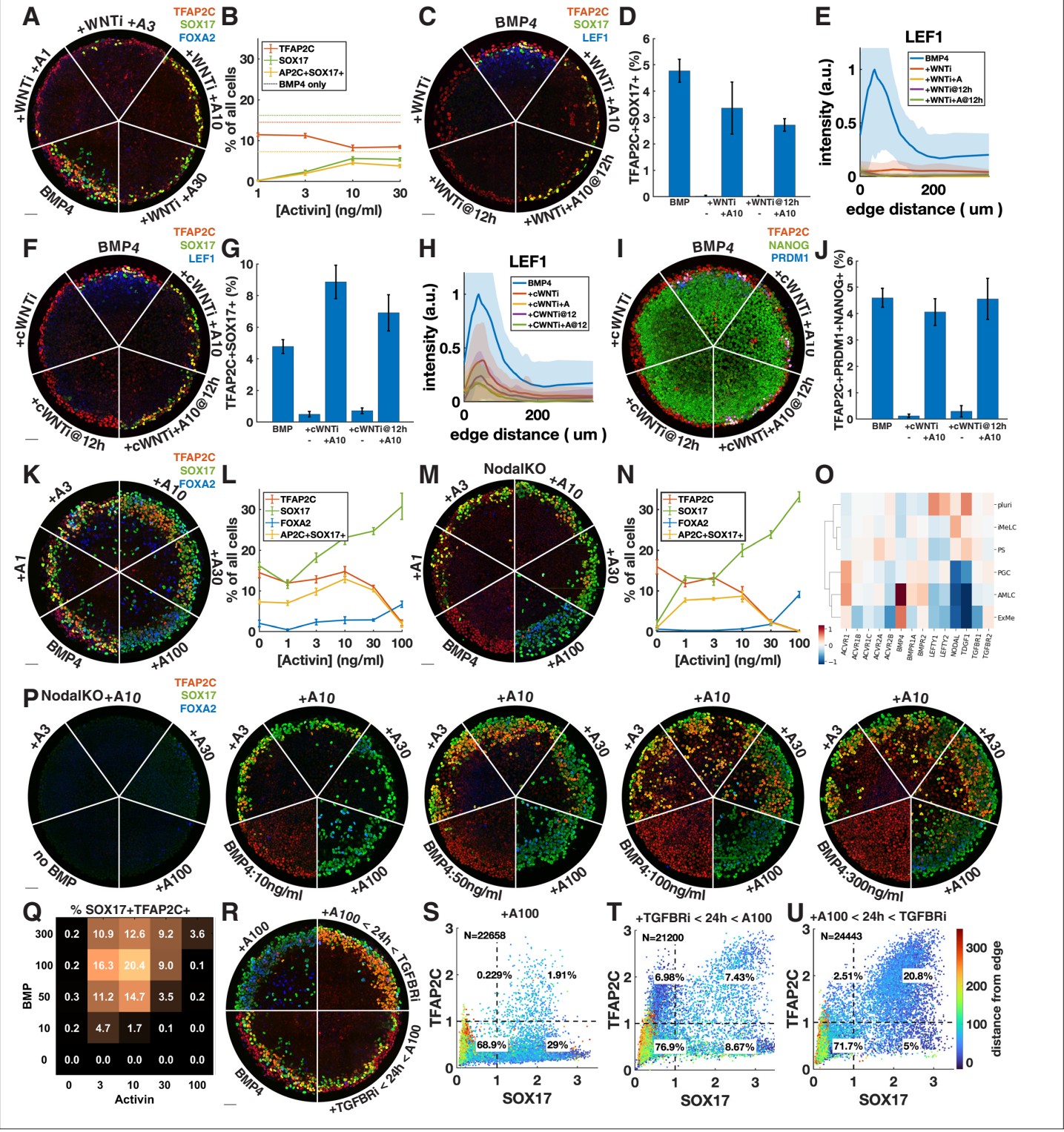

**Figure 4.** Exogenous Activin rescues primordial germ cells (PGCs) in the absence of endogenous Wnt or Nodal in a dose- and time-dependent manner. (**A, B**) Wnt inhibition (WNTi, IWP2, 5 μM) with different doses of Activin, for example, A3 = 3 ng/ml Activin. (**C–E**) Activin rescue of WNTi at 0 hr vs. 12 hr with LEF1 staining. (**F–H**) Like (**C–E**) but with canonical WNT inhibitor (cWNTi, IWR-1, 50 μM). (**I, J**) Like (**F–H**) stained for additional PGC markers. (**K–N**) Effect of treatment with Activin in WT and NodalKO cells on expression of PGC and endoderm markers. (**O**) Differential expression from scRNA-seq for Nodal and BMP receptors, as well as Nodal, BMP, and Lefty. (**P, Q**) Effect of Activin treatment on PGC differentiation of NodalKO cells for different doses

*Figure 4 continued on next page*

*Figure 4 continued*

of BMP and quantification. (**R–U**) Effect of 100 ng/ml Activin for 42 hr, only during the first 24 hr, or only after 24 hr. Images of each channel separately are shown in *Figure 3—figure supplement 1*. Scale bar 50 μm.

The online version of this article includes the following figure supplement(s) for figure 4:

**Figure supplement 1.** Additional images of primordial germ cell (PGC) rescue by Activin after WNT inhibition.

**Figure supplement 2.** Additional images of the effect of Activin timing and dose on primordial germ cell (PGC) specification.

We then asked whether Nodal signaling might generally be the limiting factor in directing TFAP2C-positive cells to PGCLC fate and treated colonies with different doses of Activin in addition to BMP4 (*Figure 4K and L*, *Figure 4—figure supplement 2A*). We found a moderate increase in PGCLCs at intermediate doses of Activin while at high doses TFAP2C was replaced by SOX17+ FOXA2+ cells. Nodal autoactivates, so it is not clear how endogenous Nodal downstream of Activin changes and contributes to its effect. We therefore repeated this experiment in NodalKO cells (*Figure 4M and N*, *Figure 4—figure supplement 2B*). We found that low doses of Activin rescue PGCLCs with numbers similar to wild-type (WT) BMP4-treated cells, while higher doses behave very similar to WT cells treated with the same dose of Activin. The similarity between NodalKO and WT cells suggests that feedback reduces the additive effect that might have been expected from Activin plus endogenous Nodal. To identify possible candidates for this feedback, we examined differential expression of Nodal, BMP, and their receptors and inhibitors in the scRNA-seq data and found severely reduced expression of the Nodal co-receptor TDGF1 in the AmLC and PGCLC clusters (*Figure 4O*). This would desensitize those cells to Nodal but still allow strong response to Activin, which does not require TDGF1. We also noticed a striking upregulation of BMP-specific type 1 and 2 receptors ACVR1, BMPR1A, and BMPR2, suggesting increased sensitivity to BMP4.

The dose-dependent effect of Activin treatment may be absolute if there are gene activation thresholds related to, for example, binding affinities of Smad2, or it may be relative to BMP, with BMP and Nodal signaling competing to activate and suppress TFAP2C. To test this, we treated NodalKO colonies with 25 different combinations of Activin and BMP doses. Although differentiation became less organized at higher doses of BMP, we found reproducible behavior with PGC induction maximal at intermediate levels of both BMP and Activin. Moreover, the effect of Activin was dependent on the level of BMP. For example, treatment with 10 ng/ml Activin significantly reduced PGC numbers at 10 ng/ml BMP, but increased PGC numbers at higher doses of BMP4, indicating that relative levels are important for fate determination. We also stained for TFAP2C, PRDM1, NANOG with similar results (*Figure 4—figure supplement 2F and G*).

Unlike endogenous Nodal, high enough exogenous Activin eliminates TFAP2C and induces strong FOXA2 expression by 42 hr. We asked whether this is due to increased signaling levels or whether it is due to changes in timing and duration. Endogenous Nodal is activated with a delay downstream of BMP and Wnt and does not reach high levels until after 24 hr (*Heemskerk et al., 2019*; *Chhabra et al., 2019*). At that point, it is possible that the cells with the highest level of BMP signaling on the outside of the colony are already committed to AmLC fate and can no longer respond to Nodal or be converted to PGCLCs. Similarly, FOXA2 expression may require longer duration Nodal signaling that is not achieved by 42 hr if signaling starts at 24 hr. To test this, we treated cells with TGFBRi for the first 24 hr followed by a high dose of Activin in the last 24 hr (*Figure 4R and T*, *Figure 4—figure supplement 2C*). Consistent with our hypothesis, and in contrast to Activin treatment for the duration of the experiment, this did not eliminate TFAP2C or induce FOXA2 but instead induced PGCLCs in numbers similar to the BMP4-only control.

Combined, these data suggest that PGC specification requires that Nodal signaling should be activated in cells expressing TFAP2C to induce SOX17 before they commit to AmLC fate, but that prolonged high-level Nodal signaling suppresses TFAP2C and activates FOXA2 to give rise to endoderm. Therefore, we predicted that a high dose of activin during only the first 24 hr would be able to convert all TFAP2C-positive cells to PGCs without inducing endoderm. Indeed, we were able to double the fraction of PGCs to about 20% by 24 hr of Activin exposure with Activin/Nodal signaling inhibited after removing Activin (*Figure 4R and U*, *Figure 4—figure supplement 2C*). As before, we repeated this experiment with NodalKO cells and obtained similar results to WT (*Figure 4—figure supplement 2D and E*).

In summary, we provide evidence that an important part of the role of Wnt in PGCLC induction is indirect by inducing Nodal, and that the balance and relative timing of the Nodal and BMP pathways play a crucial role in inducing TFAP2C and SOX17 to specify PGCs.

## Control of colony size dramatically improves PGCLC differentiation efficiency

We observed that under most conditions TFAP2C+ SOX17+ cells are confined to a ring 100 μm or less in size on the edge of the colony that accounts for at most 30% of the cells. Response to exogenous BMP and Activin is well understood as an 'edge effect,' excluded from the center by receptor localization and inhibitor production (*Etoc et al., 2016*). Therefore, we hypothesized that by reducing colony size – thereby enhancing the total proportion of cells within a colony that are in contact with the edge – we would be able to induce higher proportions of cells to express SOX17 or TFAP2C and get much larger fractions of PGCLC induction. To test this, we differentiated colonies ranging from 300 μm to 100 μm in diameter (*Figure 5A–C*). We observed increasing fractions of PGCLCs as the diameter decreases, reaching a maximum for 100 μm colonies at about 50% SOX17+ TFAP2C+ (*Figure 5D*) or TFAP2C+ PRDM1+ NANOG+ (*Figure 5E and F*, *Figure 5—figure supplement 1A*). Moreover, PS-like cells as marked by high EOMES were eliminated in 100 μm colonies and nearly all cells expressed TFAP2C, suggesting that the non-PGCLCs are AmLCs. Since complex current protocols yield 20–30% PGCLCs, it is surprising that BMP4 treatment combined with controlled colony geometry alone would yield 50% PGCLCs. We asked if the yield would increase further by pre-differentiating cells to iMeLC with 12 hr of Wnt and Nodal activation as is done in current protocols or by treating with Activin for the first 24 hr as we did in *Figure 4*. Indeed, the fraction of PGCLCs increased to 70% by pre-differentiation (*Figure 5G and H*, *Figure 5—figure supplement 1A*). Treatment with Activin did not have the same effect as for large colonies (*Figure 4R*) and only slightly increased the number of PGCLCs in small colonies (*Figure 5—figure supplement 1A–C*). We repeated these experiments with a different cell line with the same result (*Figure 5—figure supplement 1D*).

Although strongly suggested by *Etoc et al., 2016*, the hypothesis that BMP signaling has a fixed range from the colony edge and therefore is high in a larger fraction of cells in smaller colonies has not been explicitly tested. Therefore, we quantified a pSMAD1 level in different-sized colonies (*Figure 5I–K*). We also stained for SMAD2/3 as a readout for Nodal signaling. This confirmed that BMP and Nodal signaling are high in a larger fraction of cells and provide a possible mechanism for the higher efficiency of PGC differentiation in small colonies. However, in the smallest colonies the number of PGCs induced at the same distance from the edge or at similar levels of pSmad1 and nuclear Smad2/3 exceeds the expectation from larger colonies, which warrants more detailed investigation of the temporal behavior of these and other pathways in the future. In summary, combining current protocols with geometric control using micropatterning more than doubles their efficiency, likely by more uniformly creating the required signaling conditions with relatively high BMP and Nodal signaling.

## A network of cross-repressive cell fates driven by BMP and Nodal signaling explains perturbations in PGCLC induction

Our experiments suggest that cells interpret the ratio of BMP and Nodal signaling levels as well as their relative timing and duration through a network of mutually repressive fates that are acquired in a switch-like manner. Although Wnt is required in combination with Nodal to induce PS-like fates, our data suggest that for PGCLC specification only low levels of WNT are required directly and its primary role is to induce Nodal, so that many of our results can be explained without considering Wnt. Moreover, differentiation of AmLCs on the colony edge does not require Nodal, and PS-like fates do not (directly) require BMP, while PGCLCs positioned in between require combined induction of both BMP and Nodal target genes, in particular TFAP2C and SOX17.

Intuitively, this would explain the unperturbed WT pattern as follows. First, higher BMP signaling on the colony edge induces TFAP2C and amnion genes faster than further inside; then, with a delay that depends on distance from the edge, Nodal signaling turns on in all cells at similar levels and induces SOX17 and PS-like genes. Cells on the outside reach a threshold to stably switch on amnion-like genes and repress other fates before SOX17 and PS-like genes are significantly induced. Cells slightly further inside reach high enough levels of both TFAP2C and SOX17 to stably switch on PGCLC genes and

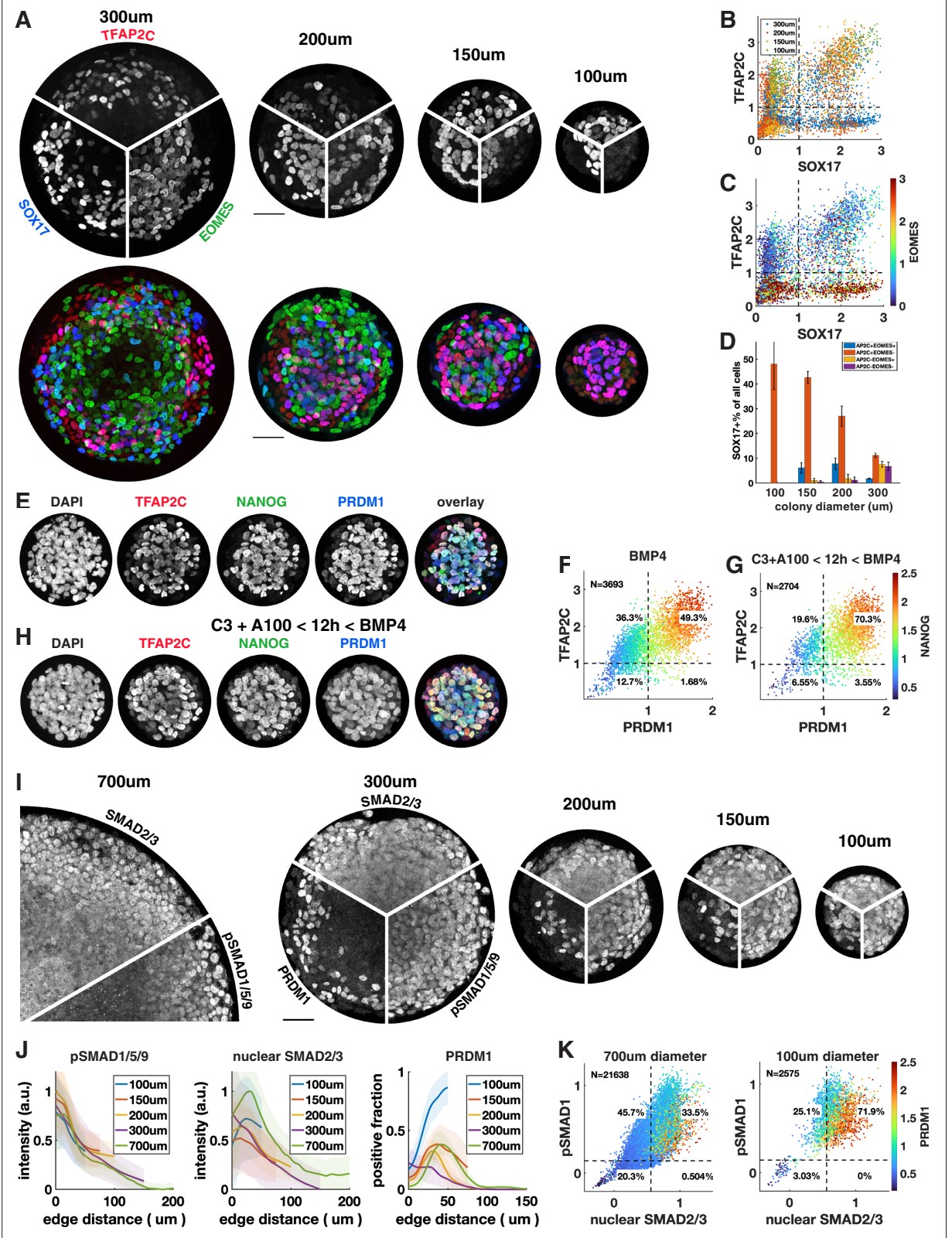

**Figure 5.** Control of colony size dramatically impacts the fraction of primordial germ cell-like cells (PGCLCs). (**A–D**) Different diameter colonies stained for TFAP2C, SOX17, EOMES at 42 hr and quantification, (**B**) SOX17 vs. TFAP2C scatterplot colored for colony size. (**C**) Same plot colored for EOMES expression. (**D**) SOX17+ subpopulations for each colony diameter. (**E–G**) 100 μm colonies differentiated with BMP only or with incipient mesoderm-

*Figure 5 continued on next page*

*Figure 5 continued*

like state (iMeLC) pre-differentiation stained for TFAP2C, NANOG, PRDM1 at 48 hr and quantification. C3 = 3 µM CHIR-99021, other notation is like in *Figure 4*. (I–K) Stainings and quantification of pSMAD1 and SMAD2/3 for different size colonies. Scale bars 50 µm.

The online version of this article includes the following figure supplement(s) for figure 5:

**Figure supplement 1.** Additional images of primordial germ cell (PGC) specification on small micropatterns.

repress other fates, while cells even further inside never significantly activate amnion genes or TFAP2C and after a longer exposure to Nodal will reach a threshold to commit to a PS-like fate. Combined induction of PS-like genes and SOX17 may specify endoderm. The effect of perturbations of Activin/ Nodal timing and duration is then naturally explained: early treatment with Activin combined with BMP can induce high enough levels of both SOX17 and TFAP2C in cells on the colony edge to make them PGCLCs before they commit to AmLC, but if the duration of Activin exposure is too long PS-like genes dominate and suppress PGCLC fate. On the other hand, Activin exposure in the second 24 hr coincides with the timing of endogenous Nodal and has little effect on fate decisions.

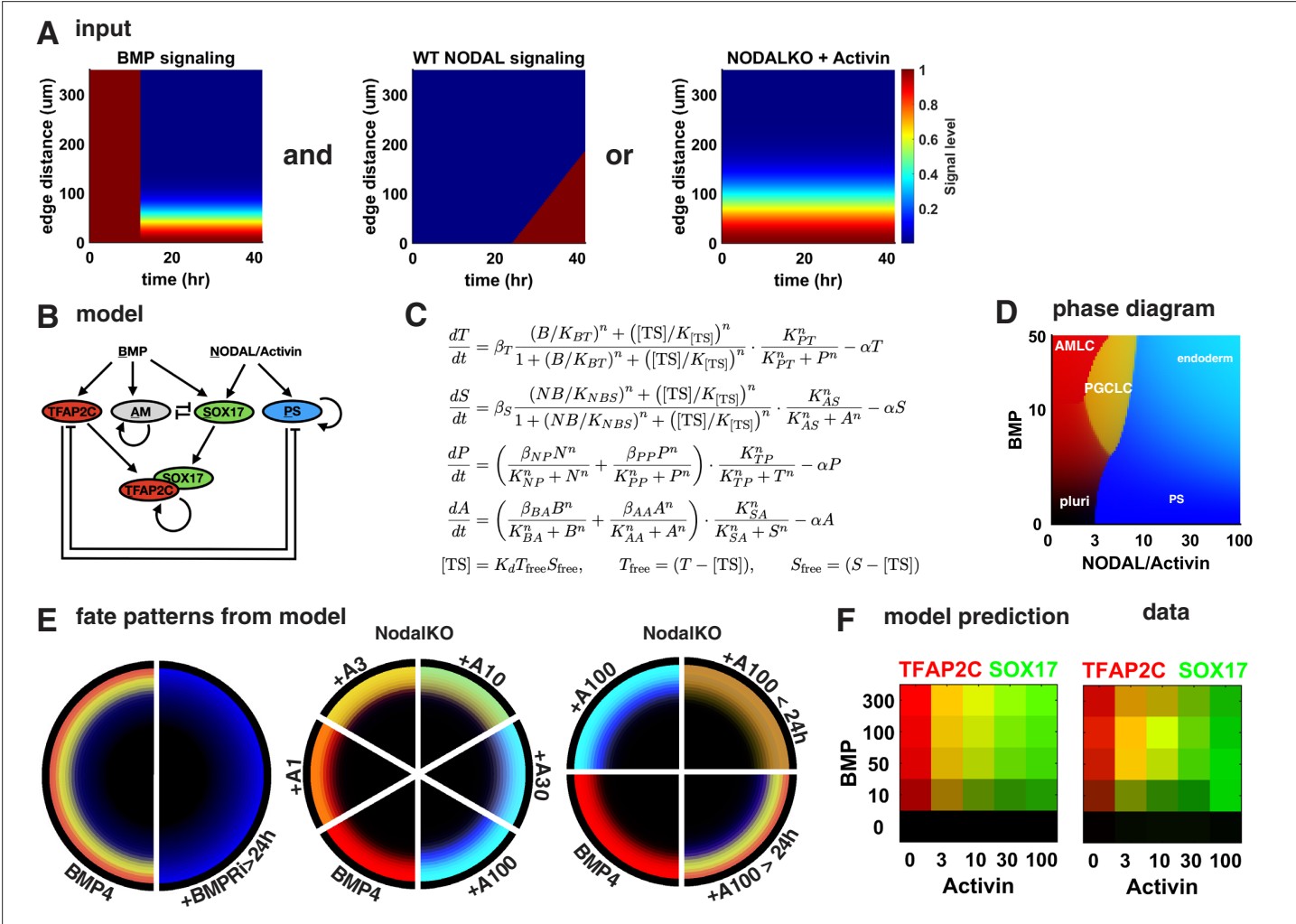

**Figure 6.** A network of cross-repressive cell fates qualitatively explains Nodal perturbations. (**A**) Input signaling profile in space and time. (**B**) Diagram of the model. (**C**) Definition of the model. (**D**) Phase diagram showing predicted expression of cell fate markers at steady state for different levels constant of BMP and Nodal activation (i.e., behavior of the cells on the very edge at late times). (**E**) Cell fate patterns predicted by the model from the input signaling profiles with different perturbations. Compare with data in *Figures 3A and 4M*, and *Figure 4—figure supplement 2D*. Colors match (**B, D**). (**F**) Predicted and measured expression of TFAP2C and SOX17 in a 100-µm-wide ring from the edge for different doses of BMP and Activin. .

The online version of this article includes the following figure supplement(s) for figure 6:

**Figure supplement 1.** Simpler mathematical models.

To test if this intuitive model holds up more rigorously and also explains our other perturbations in BMP and Nodal signaling, we identified a minimal mathematical model for the GRN specifying PGCs downstream of previously determined BMP, Nodal, and Activin signaling profiles. Previous work showed that after initial uniform activation BMP signaling is restricted to a stable gradient from the colony edge and that a region of high endogenous Nodal signaling expands into the colony from the edge at constant velocity starting around 24 hr ( *Heemskerk et al., 2019*, *Figure 6A*). Like BMP, the response to exogenous Activin forms a gradient from the edge. For simplicity, we did not include Wnt in our model because our observations appear to at least be qualitatively explained without Wnt. We also did not include FGF and its perturbations since we lack data on the spatiotemporal profile of FGF signaling. Moreover, we modeled PGCLCs as TFAP2C+ SOX17+ and treated SOX17 and PRDM1 as interchangeable in terms of the observations explained by the model, knowing that PRDM1 is downstream of SOX17. For Activin treatment, we chose to model NodalKO cells since the combined Nodal and Activin signaling is unknown and experiments suggest a feedback that results in only minor differences between WT and NodalKO. Specific choices made in the construction of the model are further detailed in Appendix 1.

The structure of the model is shown in *Figure 6B and C*, and the cell fate markers induced by constant levels of Nodal or BMP signaling after reaching steady state are shown in the phase diagram in *Figure 6D*. After fitting this model to data from *Figure 4K, M, and R*, the model is able to qualitatively reproduce the expression patterns for these conditions (*Figure 6E*). We also tested various simpler models, for example, lacking the competition between PGCLCs and the neighboring AmLC and PS-like fates and found that these are not able to fit the data. We then challenged our model to predict the effect of different BMP doses in *Figure 4P* to which we did not fit. Because our model does not simulate individual cells, we compared mean expression within 100 µm from the edge rather than % PGCs and found good agreement between model and data (*Figure 6F*). In summary, we find that the mathematical model supports our interpretation of the data.

## Discussion

In this study, we have shown that PGCs form in a very reproducible manner in BMP4-treated micropatterned hPSCs and are part of a stereotypic spatial organization that positions them between extraembryonic cells that may be amnion-like and cells expressing PS markers, similar to their location in vivo. Because they are close to the colony edge, we used micropatterning to create small colonies and were able to get much greater fractions of PGCLC differentiation than previously described. This potentially also explains the advantage of using ROCK inhibitor in PGCLC differentiation noted by some groups (*Sebastiano et al., 2021*) since that keeps cells from forming densely packed colonies and makes the majority of cells behave like the colony edge. However, micropatterning provides a more controlled way to achieve this effect. By incorporating 12 hr of iMeLC differentiation, we were able to achieve 70% efficiency compared to 10–30% described in the literature. It is possible that further protocol optimization of micropatterned differentiation could further increase the yield.

In a major advance, a microfluidics-based stem cell model of human gastrulation was recently shown to give rise to hPGCLCs and was used to study the transcriptome of hPGCs (*Zheng et al., 2019*; *Chen et al., 2019*; *Yang et al., 2021*). However, quantitative studies of the signaling dynamics underlying specification of cell populations have mostly been performed in micropatterned hPSCs due to the simplicity of the system (*Heemskerk, 2020*). The quasi-two-dimensional system provides optimal conditions for quantitative microscopy. In addition, micropatterned substrates are easy to make or purchase.

It was a surprise that we found no clear definitive endoderm (DE) population at 42 hr and that the majority of SOX17+ cells that were originally thought to be endoderm are PGCLCs. However, we found that endoderm is present at 48 hr and continues to increase until 72 hr. It is possible that the SOX17+ TFAP2C- cells at 42 hr will continue to differentiate to DE, which would require lineage tracing, but neither IF nor scRNA-seq shows expression of DE markers like FOXA2 or HEX at 42 hr. When endoderm arises, it localizes close to the PGCs and is still in a location where high BMP is expected. Therefore, the puzzle of endoderm localization in micropatterned hPSC colonies is not fully resolved by our study. However, Nodal and BMP have not been measured after 48 hr, so it is possible that BMP signaling is excluded from this region at later times. A different study performed scRNA-seq at 44 hr (*Minn et al., 2020*) and found an endodermal and PGC population. This may

capture the earliest endoderm formation we observed or reflect subtle differences in timing or differentiation potential that depend on the cell line. Our data does show some variation between cell lines (*Figure 1—figure supplement 3*). Differences in timing may also be due to details of the protocol, such as the total media volume or small differences in the initial cell density.

Our earlier work showed that endogenous Nodal does not form static gradients but moves into the colony like a wavefront with constant velocity, arguing against the classic model of pattern formation by concentration thresholds (*Heemskerk et al., 2019*). Moreover, we found that response to Activin and Nodal is adaptive and that gene response depends in part on signal rate of change. Here, we have observed both duration-dependent and dose-dependent PGCLC specification by Activin and Nodal. These findings are all consistent with our model in which gene expression depends on integrated signaling activity that could increase either through concentration, rate of concentration change, or duration. The integrated signaling over time is then interpreted by a GRN to make cell fate decisions.

Although to our knowledge the interactions in our model for the GRN are consistent with the literature, there are several variations possible at the level of the model, and several molecular mechanisms that could be responsible for the behavior of the same model (see Supplementary material text). For example, instead of directly, BMP could activate SOX17 indirectly through TFAP2C as suggested by *Chen et al., 2019*. Future work will refine this model to make more accurate predictions for the markers of interest and move towards quantitative predictions of PGCLC specification. Future refinements of the model will also have to include the activity of the FGF and Wnt pathways. While a model involving only BMP and Nodal explained many of our observations, it cannot explain the effect of Wnt and FGF inhibition or quantitatively explain what separates PGCs from neighboring cells with similar BMP and Nodal signaling.

It will also be important to relate our results to in vivo development in more detail. One question is the origin of PGCs, which in cynomolgus monkeys were found to be the amnion around day 11 (*Sasaki et al., 2016*). That in vivo work defined the amnion based on its location: facing the trophoblast, whereas the epiblast faces the hypoblast. However, a clear molecular signature of amnion was not found until day 14 (*Yang et al., 2021*). Therefore, it is possible that early amnion consists of pluripotent cells exposed to BMP and other signals from the trophoblast, which then gives rise to both committed amnion and PGCs after downregulation of SOX2. This would be consistent with our model where the outer cells exposed to high BMP give rise to amnion-like and PGC-like cells. Another question in relation to in vivo data is the precise role of BMP. In contrast to our data, which shows PGC markers correlate with intermediate to high levels of BMP signaling, it was recently found that in the mouse BMP signaling is lower in PGCs relative to neighboring cells at E7.5 and that pSmad1/5/9 signaling does not appear to be cell-autonomously required (*Senft et al., 2019*; *Morgani and Hadjantonakis, 2021*). Differences in BMP signaling between the systems could be due to developmental progression since BMP signaling in murine (pre-)PGCs gradually goes down from E5.5 to E7.5, and PGCs in our system may downregulate BMP signaling as they mature. Moreover, it is possible that the dependence on BMP we demonstrated in our system is indirect through the amnion-like cells, but not through other cell types since those are not present on small micropatterns. However, given that PGCs in primates arise through a different GRN, at a different time, surrounded by different extraembryonic tissues, it is also plausible that BMP signaling in mouse and primate PGCs is qualitatively different.

While the focus in human PGCLC differentiation has typically been on BMP and Wnt, we showed that a major part of the role of Wnt is to induce Nodal and that the effect of Wnt inhibition on hPGCLC specification can be rescued by exogenous Activin. This further highlights the complex feedback between the paracrine signaling pathways that make it hard to directly interpret the effect of a signaling perturbation, and therefore the need for a quantitative approach. By establishing a highly efficient and reproducible differentiation platform and revealing how timing, duration, and dose of Activin/Nodal signaling affect hPGCLC specification, we have laid the foundation for future quantitative investigations of the interplay between different signaling pathways during PGCLC induction, and the downstream GRN that interprets these signals to determine fate.

**Table 1.** Cell signaling reagents.

| Reagent | Nickname | Vendor, catalog # | Dose | Function |
|---|---|---|---|---|
| rhBMP4 | BMP4 | R&D Systems, #314BP/CF | See figures | Activate BMP pathway |
| rhActivin | A | R&D Systems, #AFL338 | See figures | Activate TGFb pathway |
| CHIR-99021 | C | Tocris, #4423 | See figures | Canonical Wnt agonist |
| IWP 2 | WNTi | Tocris, #3533 | 5 µM unless stated otherwise | Block Wnt secretion |
| IWR-1 | cWNTi | Thermo Fisher, 50-101-4191 | 50 µM unless stated otherwise | Block canonical Wnt signaling |
| LDN-193189 | BMPRi | MedChemExpress, # HY-12071 | 250 nM | Block BMP signaling |
| SB-431542 | TGFBRi | Apexbio, #A8249 | 10 µM | Block TGFb signaling |
| PD-0325901 | MEKi | ESIBIO, #ST10009 | 5 µM | Block MEK signaling |
| PD-173074 | FGFRi | MedChemExpress, #HY-10321 | 1 µM | Block FGF signaling |

## Materials and methods

### Replicates, sample sizes, and error bars

All experiments were performed at least twice. Quantification was performed on four or more colonies per condition in each experiment. Numbers of cells in IF analysis are shown in each scatterplot. Error bars in quantitative image analysis represent standard deviation over colonies unless otherwise stated. Error bars on qRT-PCR data are over technical triplicates from the representative biological sample set.

### Cell lines

The cell lines used were the embryonic stem cell line ESI017 (XX), and the induced pluripotent stem cell lines PGP1 (XY), WTC11 (XY), MR30 (XX). The identity of these cells as pluripotent stem cells was confirmed by staining of pluripotency markers OCT3/4, SOX2, NANOG. All cells were routinely tested for mycoplasma contamination, and negative results were recorded.

### Pluripotent stem cell culture and differentiation

Pluripotent stem cells were cultured in the chemically defined mTeSR1 media (StemCell Technologies) on Cultrex (R&D Systems)-coated tissue culture plates. mTeSR1 contains TGF$\beta$ (0.6 ng/ml) and FGF2 (100 ng/ml). Whole-colony routine passaging was done using L7 (*Nie et al., 2014*), and single-cell suspension for seeding experiments was generated using Accutase. For micropatterned colonies, we followed *Deglincerti et al., 2016*. In short, cells were seeded as a single-cell suspension onto laminin-coated micropatterns in mTeSR1 with ROCK inhibitor. Two hours after seeding, media was changed for mTeSR1 without ROCK inhibitor and with BMP4. Unless stated otherwise, BMP4 treatment was done with 50 ng/ml. All experiments were done in micropatterned 18-well Ibidi slides made using the protocol (*Azioune et al., 2009*). All colonies were 700 µm diameter unless stated otherwise. Reagents to modify signaling during pattern formation are listed in *Table 1*.

### Imaging and image analysis

Imaging was done on an Andor Dragonfly/Leica DMI8 spinning disk confocal microscope with a ×40, NA 1.1 water objective. Nuclei were segmented in individual z-slices based on DAPI staining using two different machine learning approaches: Ilastik (*Sommer et al., 2011*) and Cellpose (*Stringer et al., 2021*). We found that Cellpose is highly accurate with segmenting the nuclei it finds, but it sometimes misses lower contrast nuclei, whereas Ilastik can be easily trained to find all nuclei but more frequently has trouble separating neighboring nuclei. Therefore, we combined the two segmentations in each z-slice giving preference to Cellpose. We start with the Cellpose segmentation and then take all pixels of the Ilastik nuclear mask that are not members of any Cellpose nuclear mask as a binary mask of all the nuclei missed by Cellpose. We remove small noise features from this mask with a morphological opening operation and identify connected components of the resulting mask as individual nuclei at each z-slice, separating merged or overlapping nuclei with a convex decomposition algorithm.

To get a 3D segmentation, we next linked segmentations in different z-slices using a linking algorithm formulated as a linear assignment problem loosely based on the particle tracking approach in *Jaqaman et al., 2008*. To link nuclei in slice zn to nuclei in slice zn + 1, we defined the cost matrix for the LAP as a block matrix of the form

$$\begin{bmatrix} A & B \\ C & A^T \end{bmatrix}.$$

A(i, j) gives the cost of linking nucleus i in frame n to nucleus j in frames n + 1, and is given by

$$A(i,j) = \begin{cases} \frac{\min(|N_{n,i}|,|N_{n+1,i}|)}{|N_{n,i} \cap N_{n+1,j}|} & if\ d(i,j) \leq d_{\max} \\ \text{Inf} & if\ d(i,j) > d_{\max} \end{cases},$$

where $N_{n,i}$ is the set of pixels in the mask of nucleus i in frame j and d(i, j) is the distance between the centroids of the two masks. That is, the cost to link two nuclei is the smaller of the sizes of the two masks divided by the size of their overlap. For efficiency, each nucleus in slice n has this cost computed only for its three nearest neighbors in slice n + 1 and vice versa, and all other costs are set to Inf (arbitrarily large, so that these links are treated as impossible). We further impose a cutoff dmax on the distance between the centroids of the two nuclei and set A(i,j) = Inf if the distance exceeds the cutoff. Finally, B and C are square diagonal matrices with all off-diagonal entries set to Inf and diagonal entries set to the 'alternative cost' 1/IoU for not linking to any other nucleus, where IoU is an intersection over union threshold set to determine the minimum ratio of overlap to nucleus area that qualifies two nuclei to be linked. If every cost along the ith row of A exceeds 1/IoU, then nucleus i in slice n will be linked to nothing, and likewise for costs along columns. This linking operation is performed sequentially across pairs of adjacent z-slices, creating chains of linked masks in different slices that are taken to correspond to a single nucleus. We additionally impose a maximum expected nuclear diameter and use the spacing between z-slices to determine the maximum number of slices that may correspond to a single nucleus. If more than this number of masks are linked together, the chain is broken into two parts by splitting it at a local minimum in the area of the nuclear mask. Since nuclei are defined across multiple z-slices, a given nucleus has a readout of average fluorescent intensity in each channel in each slice. For each channel, we take the maximum across z-slices as the value for that nucleus as it should correspond to the readout in which the nucleus was most nearly in focus.

Using the resulting segmentation, we extracted mean intensities for each of the stained markers in each nucleus. For further analysis, the single-cell expression data obtained this way was log(1 + x) transformed similar to what is common for scRNA-seq analysis for several reasons including reduction of the effect of outliers on the analysis. We separated population by thresholds in each marker, which while not perfect performed better than more advanced clustering methods. To determine a threshold between cells expressing or not expressing a marker, we fitted a Gaussian mixture model

**Table 2.** Primary antibodies used for immunostaining.

| Protein | Species | Dilution | Catalog # | Vendor |
|---|---|---|---|---|
| ISL1 | Mouse | 1:200 | 39.4D5 | DSHB |
| SOX2 | Rabbit | 1:200 | 3579S | Cell Signaling Technology |
| TBXT (BRA) | Goat | 1:300 | AF2085 | R&D Systems |
| PRDM1 (BLIMP1) | Rat | 1:50 | SC-47732 | Santa Cruz Biotechnology |
| SOX17 | Goat | 1:200 | AF1924 | R&D Systems |
| TFAP2C | Mouse | 1:150 | SC-12762 | Santa Cruz Biotechnology |
| NANOG | Goat | 1:100 | AF1997 | R&D Systems |
| EOMES (TBR2) | Rabbit | 1:500 | AB23345 | Abcam |
| POU5F1 | Mouse | 1:400 | 611,202 | BD Biosciences |
| LEF1 | Rabbit | 1:200 | C12A5 | Cell Signaling Technology |

**Table 3.** Secondary antibodies.

| Protein | Species | Dilution | Catalog # | Vendor |
|---|---|---|---|---|
| Alexa Fluor 647 anti-goat | Donkey IgG | 1:500 | A21447 | Thermo Fisher Scientific |
| Alexa Fluor 555 anti-goat | Donkey IgG | 1:500 | A21432 | Thermo Fisher Scientific |
| Alexa Fluor 488 anti-mouse | Donkey IgG | 1:500 | A21202 | Thermo Fisher Scientific |
| Alexa Fluor 647 anti-rat | Whole IgG | 1:500 | 112-605-167 | Jackson ImmunoResearch |
| Alexa Fluor 647 anti-rabbit | Donkey IgG | 1:500 | A31573 | Thermo Fisher Scientific |
| Alexa Fluor 555 anti-rabbit | Donkey IgG | 1:500 | A31572 | Thermo Fisher Scientific |

to the expression data of each gene separately, which worked better than fitting it to the combined gene expression due to the clusters not being sufficiently Gaussian in two or three dimensions. The number of Gaussians was determined automatically using the Bayesian information criterion, and the positive cells were taken to be those belonging to the Gaussian with the highest mean. This generally produced good results but, in some cases, did require manual fine-tuning based on the scatterplot (thresholds shown in all scatterplots). The data were rescaled so that $\log(1 + x\_thresh) = 1$ for visualization in scatterplots. All codes are available on github.com/idse/PGCs.

## Immunostaining

Coverslips were rinsed with PBS, fixed for 20 min in 4% paraformaldehyde, rinsed twice with PBS, and blocked for 30 min at room temperature with 3% donkey serum and 0.1% Triton X-100 in 1× PBS. After blocking, cells were incubated with primary antibodies at 4°C overnight, followed by three washes in PBST (PBS with 0.1% Tween 20). They were then incubated with secondary antibodies and DAPI for 30 min at room temperature and washed twice in PBST at room temperature. In some cases, repeated stainings were done following the protocol from *Gut et al., 2018*. Antibodies can be found in *Tables 2 and 3*.

### qPCR

For qPCR experiments, ESI017 cells were grown in 24-well plates or 18-well Ibidi slides. For EOMES response in *Figure 4C*, all treatments were done by taking part of the media from each well to dilute treatment reagents that were then added back to the well in order to prevent effects of adding fresh media. RNA was extracted using Ambion RNAqueous-Micro Total RNA Isolation Kit, and cDNA synthesis was performed with Invitrogen Super- Script Vilo cDNA Synthesis Kit according to the manufacturer's instructions. Measurements were performed in technical triplicate with SYBR green; primers are given in *Table 4*. GAPDH was used for normalization. In all cases, at least two biological replicates were performed and showed similar results.

### scRNA-seq

Cells were collected using accutase and resuspended in ice-cold PBS. Single-cell RNA-sequencing was performed by the University of Michigan Advanced Genomics Core. Cells were barcoded using the 10X Genomics Chromium system (part numbers 1000268, 1000120, 1000215). For quality control,

**Table 4.** qPCR primers.

| GAPDH | ACAACTTTGGTATCGTGGAAGG | GCCATCACGCCACAGTTTC |
|---|---|---|
| SOX17 | GTGGACCGCACGGAATTTG | GGAGATTCACACCGGAGTCA |
| NANOS3 | CTTTGACCTGTGGACAGATTACC | GCCTGGTTTCAGGACCCTC |
| DPPA3 | TTAATCCAACCTACATCCCAGGG | AGGGGAAACAGATTCGCTACTA |
| DDX4 | TTGTTGCTGTTGGACAAGTGGGTG | GCAACAAGAACTGGGCACTTTCCA |
| EOMES | CGCCACCAAACTGAGATGAT | CACATTGTAGTGGGCAGTGG |
| PRDM1 | CTACCCTTATCCCGGAGAGC | GGACATTCTTTGGGCAGAGT |

cDNA was quantified by Qubit High Sensitivity DNA assay and Agilent TapeStation. Sequencing was performed on the Illumina NovaSeq 6000 with NovaSeq S4 flowcell and Control Software version 1.7.0. Reads were aligned using cellranger-4.0.0 with the GRCh38 reference. Further processing was done in Python using the scprep package, and the script used for all analyses that include all parameters is included as a supplement. After filtering for library size to exclude empty droplets and duplets, 4254 cells were left. After excluding outliers for mitochondrial gene expression and excluding genes that were expressed in fewer than 50 cells, we were left with 4095 cells and 16,151 genes. The data was then transformed using a sqrt, which has a similar effect as the commonly used $\log(1 + x)$ transformation without the arbitrary pseudocount to avoid singular behavior at zero. Rather than regress out various factors like cell cycle or pseudogenes that were not of interest or may confound analysis and lead to misinterpretation (*Chhabra and Warmflash, 2021*), we made a list of developmental genes of interest (*Supplementary file 1*) that we used for visualization and clustering. We found both visualization and clustering to be more reliable and more informative this way and found our results to be very stable to adding genes or to removing genes from this list. Dimensional reduction for visualization was performed using PHATE, which preserves the global structure, that is, lineage structure of the data better than UMAP without compromising local structure. For visualizing gene expression on PHATE plots and visualizing gene relationships using DREMI, we first performed denoising using MAGIC. Other analysis such as differential expression was performed on the full data. Data was scaled to zero mean and unit variance before performing differential expression analysis using Earth Mover's Distance.

## Acknowledgements

We thank Aryeh Warmflash, Sue Hammoud, and Craig Johnson for discussions. We also thank Patrick Oakes for help with troubleshooting micropatterning. NodalKO cells were a gift from Aryeh Warmflash. This work was supported by the Branco Weiss Fellowship – Society in Science, the University of Michigan Pioneer Postdoctoral Fellowship, the University of Michigan, and the National Institute of General Medical Sciences.

## Additional information

### Funding

| Funder | Grant reference number | Author |
|---|---|---|
| University of Michigan Medical School | startup | Kyoung Jo<br>Seth Teague<br>Bohan Chen<br>Hina Aftab Khan<br>Emily Freeburne<br>Hunter Li<br>Bolin Li<br>Ran Ran<br>Idse Heemskerk |
| ETH Zürich Foundation | Branco Weiss Fellowship | Hina Aftab Khan<br>Idse Heemskerk |
| National Institute of General Medical Sciences | R35 GM138346 | Seth Teague<br>Bohan Chen |
| University of Michigan Medical School | Pioneer Fellowship | Kyoung Jo |

The funders had no role in study design, data collection and interpretation, or the decision to submit the work for publication.

### Author contributions

Kyoung Jo, Data curation, Investigation, Methodology, Supervision, Visualization, Writing – original draft, Writing – review and editing; Seth Teague, Formal analysis, Investigation, Methodology, Software, Visualization, Writing – original draft, Writing – review and editing; Bohan Chen, Formal analysis,

Investigation, Writing – original draft, Writing – review and editing; Hina Aftab Khan, Emily Freeburne, Hunter Li, Bolin Li, Ran Ran, Investigation; Jason R Spence, Supervision, Writing – review and editing; Idse Heemskerk, Conceptualization, Data curation, Formal analysis, Funding acquisition, Investigation, Methodology, Project administration, Software, Supervision, Visualization, Writing – original draft, Writing – review and editing

## Author ORCIDs
Bohan Chen http://orcid.org/0000-0002-9781-2982
Emily Freeburne http://orcid.org/0000-0003-0344-577X
Jason R Spence http://orcid.org/0000-0001-7869-3992
Idse Heemskerk http://orcid.org/0000-0002-8861-7712

## Decision letter and Author response
Decision letter https://doi.org/10.7554/eLife.72811.sa1
Author response https://doi.org/10.7554/eLife.72811.sa2

# Additional files

## Supplementary files
- Supplementary file 1. List of gastrulation genes used for visualization and clustering.
- Supplementary file 2. Most differentially expressed genes in clusters found in scRNA-seq data.
- Supplementary file 3. Most differentially expressed genes from *Supplementary file 1* in clusters found in scRNA-seq data.
- Transparent reporting form

## Data availability
All code for data analysis and model simulations is available on (https://github.com/idse/PGCs, copy archived at swh:1:rev:9c52edf907e9d4251ada6b85a99f4edc13784eeb) scRNA-seq data have been deposited in GEO under accession number GSE182057.

The following dataset was generated:

| Author(s) | Year | Dataset title | Dataset URL | Database and Identifier |
|---|---|---|---|---|
| Jo K, Heemskerk I | 2021 | scRNA-seq of BMP-treated micropatterned hPSCs after 42h | http://www.ncbi.nlm.nih.gov/geo/query/acc.cgi?acc=GSE182057 | NCBI Gene Expression Omnibus, GSE182057 |

The following previously published dataset was used:

| Author(s) | Year | Dataset title | Dataset URL | Database and Identifier |
|---|---|---|---|---|
| Tyser et al | 2020 | Human gastrula | http://human-gastrula.net/ | Human Gastrulation Data, human-gastrula |

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

## Appendix 1

## ODE model for hPGCLC specification

### Desired qualitative behavior

We aimed to identify the minimal mathematical model that recapitulates the following qualitative features of PGCLC specification in response to BMP and NODAL/Activin signaling:

1. TFAP2C does not depend on NODAL but is eliminated by adding BMPRi after 24 hr, while SOX17 does require NODAL (*Figure 3A and P*).
2. PGCs are stable and express higher TFAP2C than TFAP2C+ SOX17- cells that disappear over time (*Figures 1 and 2*).
3. TFAP2C and SOX17 are expressed in concentric rings that generally extend no more than 100 μm from the colony edge (*Figure 1A*).
4. High Activin throughout differentiation eliminates TFAP2C expression and gives rise to meso-derm and endoderm while lower doses have only a small effect in WT cells and rescue WT levels in NodalKO (*Figure 4K–N*).
5. High Activin during only the first 24 hr results in SOX17 co-expression in all TFAP2C+ cells, while high Activin only after the first 24 hr leads to much less SOX17 with outer cells remaining TFAP2C+/SOX17- (*Figure 4R–U*, *Figure 4—figure supplement 2C and D*).

### GRN construction

To explain this set of observations, we formulated a hypothetical GRN integrating NODAL/Activin and BMP signaling to decide between amnion-like, mesendodermal, and primordial germ cell-like fates. For simplicity, all activating and inhibitory interactions are taken to act transcriptionally rather than as post-transcriptional or post-translational interactions.

To explain the first observation, and in agreement with *Kojima et al., 2017*, we take TFAP2C to be directly activated by BMP signaling and take SOX17 to be activated by Activin/NODAL signaling. SOX17 induction also depends on NODAL indirectly through EOMES, but we do not include this complication in our model for simplicity.

The second observation suggests that expression of TFAP2C and SOX17 is maintained by positive feedback, but that this positive autoregulation functions only when both genes are present. Several mechanisms could explain this behavior; for instance, SOX17 and TFAP2C could bind independently at different regions of their promoters, but stimulate transcription only when both are present. Alternatively, SOX17 and TFAP2C may only regulate their own transcription in complex with one another. Because SOX proteins are known to complex with a partner transcription factor (TF) for their action (*Kamachi and Kondoh, 2013*), we favor the latter explanation and model SOX17 and TFAP2C as forming a dimer to upregulate their own expression, but this is not essential for the behavior of the model. This results in a stable population of SOX17+/TFAP2C+ cells, but no stable SOX17-/TFAP2C+ population.

The third observation requires a mechanism to restrict expression of both genes to a ring near the colony edge in all cases. It has been shown (*Etoc et al., 2016*; *Heemskerk et al., 2019*; *Chhabra et al., 2019*) that in BMP4-driven differentiation of micropatterned hPSC colonies BMP signaling is initially uniformly high throughout the colony, before becoming restricted to the colony edge with a sharp gradient towards the colony center after about 12 hr due to receptor relocalization and production of diffusible BMP inhibitors (*Figure 6A*). We expect that TFAP2C expression is restricted to near the colony edge because this is the only region of the colony that experiences sustained high BMP signaling. However, in WT micropatterned colonies, it has been observed (*Heemskerk et al., 2019*; *Chhabra et al., 2019*) that a wave of high NODAL signaling begins at the colony edge at about 24 hr and moves inward toward the colony center at a constant velocity, reaching most of the colony by 42 hr, so that the range of SOX17 expression cannot be explained only by the range of NODAL signaling (*Figure 6A*). We propose that SOX17 requires both NODAL and BMP signaling at relatively low levels in order to restrict expression from the colony center. An alternative explanation could be that instead of relying directly on BMP, SOX17 relies on induction by a BMP target TF such as TFAP2C as suggested by *Chen et al., 2019* or GATA3 as suggested by *Kojima et al., 2021*.

The center of WT BMP4-treated micropatterns experiences high BMP signaling only during the first 12 hr of differentiation and high NODAL signaling only after the first 24, preventing induction of SOX17 in this region in our model. Because SOX17 is absent in the colony center, any initial expression of TFAP2C in response to high BMP signaling in this region is not sustained. Also note that it has been observed (*Etoc et al., 2016*) that densely packed hPSCs have reduced sensitivity to (apical) Activin treatment due to receptor relocalization, so that Activin-treated colonies form a gradient of Smad2/3 activity that is highest at the colony edge, with low signaling in the center. However, we expect the gradient from the colony edge to be less steep than for BMP since, unlike for BMP, there is no evidence of a spatial profile of signaling inhibitors that is highest in the colony center (*Figure 6A*).

Observation four suggests that Activin can inhibit TFAP2C expression, but only at the highest doses and for treatment that is sustained for >24 hr. A simple mechanism that could explain this observation is that Activin stimulates the production of another TF that inhibits TFAP2C but only after it accumulates to a sufficiently high level. Furthermore, for the highest Activin rescue doses in both WT and NODALKO cells, there remains a ring of SOX17 expression of similar size, but most of these cells co-express the definitive endoderm gene FOXA2 instead of the PGC marker TFAP2C, indicating that differentiation is switched to mesendoderm. For this reason, we take the proposed inhibitory TF acting on TFAP2C to be a specifier of primitive streak and refer to it generically as PS. We further observed that inhibition of TFAP2C, once it does happen, appears to be complete, so that high sustained Activin fully represses TFAP2C, while lower or transient treatment has little or no effect. To explain both the observed delay before inhibition begins, and its switch-like nature, we propose that PS initially slowly accumulates in response to NODAL/Activin signaling before reaching a threshold for autoactivation, after which it rapidly stimulates its own production to a high level. A similar separation of production into an initial slow regime followed by fast autoactivation was suggested by the transcriptional dynamics of a subset of BMP4 and Activin target genes in *Heemskerk et al., 2019*, including the primitive streak markers TBXT and EOMES. An alternative mechanism to explain this delay and switch-like activation could be an additional TF upstream of PS that activates it in a coherent feedforward loop with Activin signaling; to keep the minimal number of genes in the network, we model only the first proposed mechanism. We additionally take TFAP2C to repress production of PS in agreement with the observation in *Chen et al., 2019* that TFAP2C inhibits differentiation to primitive streak derivatives, and to explain the observation that higher BMP doses require a higher Activin dose to repress PGCLC fate (*Figure 4P and Q*).

The final observation suggests that sustained high BMP signaling at the outermost edge of the colony in the absence of NODAL/Activin signaling causes cells to commit to an amnion-like fate that prevents later induction of SOX17 in response to Activin or NODAL. To impose this behavior, we included an amnion-specific TF, referred to as AM, with a formulation similar to that of PS: slow accumulation in response to high BMP signaling before reaching an autoactivation threshold and robustly upregulating its own production, and we also take AM and SOX17 to be mutually inhibitory. Because we see that the outermost cells in the colony become TFAP2C+/SOX17- and a ring inset from these becomes TFAP2C+/SOX17+, AM autoactivation must occur early enough to prevent induction of SOX17 only with the highest levels of sustained BMP at the colony edge, while TFAP2C and SOX17 can respond to intermediate levels farther down the BMP gradient. An alternative and possibly redundant mechanism to prevent PGCLC differentiation on the colony edge could be BMP-dependent desensitization to NODAL/Activin signaling via downregulation of TGFβ receptors and upregulation of NODAL/Activin inhibitors; for simplicity, we did not include this in the model.

We tested systematically if the model identified this way could be further simplified by taking parts out and testing the predicted cell fate patterns for each perturbation (*Figure 6—figure supplement 1*). Each of the simplified models failed to correctly predict part of the patterns. A graphical sketch of the GRN is included in *Figure 6* and *Figure 6—figure supplement 1*.

## ODE model

Denote NODAL/Activin as $N$, BMP as $B$, TFAP2C as $T$, SOX17 as $S$, and the primitive streak and amnion transcription factors PS and AM as $P$ and $A$, respectively. Degradation rates are denoted $\alpha$, maximal production rates are denoted $\beta$, and $K_{XY}$ gives the activation or inhibition threshold for $X$ acting on $Y$. Each transcription factor is assumed to positively regulate its own production, and all input functions are taken as Hill functions with coefficient $n$. We also take the combined dilution and

degradation rate $\alpha$ for each gene to be approximately equal. In agreement with the simple GRN sketched above, we can write the following system of ordinary differential equations (ODEs):

$$\frac{dT}{dt} = \beta_T \frac{(B/K_{BT})^n + ([TS]/K_{[TS]})^n}{1 + (B/K_{BT})^n + ([TS]/K_{[TS]})^n} \cdot \frac{K_{PT}^n}{K_{PT}^n + P^n} - \alpha T, \tag{1}$$

$$\frac{dS}{dt} = \beta_S \frac{(NB/K_{NBS})^n + ([TS]/K_{[TS]})^n}{1 + (NB/K_{NBS})^n + ([TS]/K_{[TS]})^n} \cdot \frac{K_{AS}^n}{K_{AS}^n + A^n} - \alpha S, \tag{2}$$

$$\frac{dP}{dt} = \left( \frac{\beta_{NP} N^n}{K_{NP}^n + N^n} + \frac{\beta_{PP} P^n}{K_{PP}^n + P^n} \right) \cdot \frac{K_{TP}^n}{K_{TP}^n + T^n} - \alpha P, \tag{3}$$

$$\frac{dA}{dt} = \left( \frac{\beta_{BA} B^n}{K_{BA}^n + B^n} + \frac{\beta_{AA} A^n}{K_{AA}^n + A^n} \right) \cdot \frac{K_{SA}^n}{K_{SA}^n + S^n} - \alpha A, \tag{4}$$

$$[TS] = K_d T_{\text{free}} S_{\text{free}}, \qquad T_{\text{free}} = T - [TS], \qquad S_{\text{free}} = S - [TS]. \tag{5}$$

The concentration of the dimer [TS] in (5) is determined by the equation

$$\frac{d[TS]}{dt} = \gamma_{[TS]} T_{\text{free}} S_{\text{free}} - \gamma_{[TS]} [TS], \tag{6}$$

where $\gamma_{TS}$ and $\gamma_{[TS]}$ are association and disassociation constants for the complex. Since complex formation happens much faster than protein production and degradation, we can take this equation to be effectively at equilibrium on the timescale of the above system of ODEs, so that

$$[TS] = K_d T_{\text{free}} S_{\text{free}}, \tag{7}$$

where $K_d = \gamma_{TS}/\gamma_{[TS]}$. We can express $T_{\text{free}}$ and $S_{\text{free}}$ in terms of the total concentration of $T$ and $S$ as $T_{\text{free}} = T - [TS]$ and $S_{\text{free}} = S - [TS]$. Substituting into (7) and rearranging yields

$$[TS]^2 - (T + S + 1/K_d)[TS] + TS = 0,$$

which has the solutions

$$[TS] = \frac{T + S + 1/K_d \pm \sqrt{(T + S + 1/K_d)^2 - 4TS}}{2}.$$

Given the additional constraint that $[TS] \leq \min(T, S)$, we can discard the larger root as it is always larger than both $T$ and $S$, so that

$$[TS] = \frac{T + S + 1/K_d - \sqrt{(T + S + 1/K_d)^2 - 4TS}}{2}.$$

Note that as $K_d$ becomes large, [TS] approaches $\min(T, S)$. When numerically evaluating the system of ODEs, the dimer concentration is set to this steady-state value depending on absolute concentrations of $T$ and $S$ at each step.

In (1) and (2), $T$ and $S$ have activation functions written such that production by either autoactivation or signaling inputs is a Hill function with coefficient $n$, but with the Hill functions combined to approximate OR logic; that is, for either $[TS] \gg K_{[TS]}$ or $B \gg K_{BT}$,

$$\beta_T \frac{(B/K_{BT})^n + ([TS]/K_{[TS]})^n}{1 + (B/K_{BT})^n + ([TS]/K_{[TS]})^n} \rightarrow \beta_T,$$

so that production of $T$ can be driven at similar levels by either $B$ or [TS]. Similarly, production of $S$ can be driven either by [TS] or by $A$ and $B$ together. The inhibition function for each gene, on the other hand, is combined multiplicatively to approximate AND logic so that sufficiently high levels of inhibitor can completely downregulate production. For instance, $T$ is produced only if ($[TS] > K_{[TS]}$ OR $B > K_{BT}$) AND ($P < K_{PT}$).

*Equations (1) and (2)* allow a relatively short exposure time to BMP and Activin to induce TFAP2C and SOX17. We wrote (3), however, according to the observation that PS turns on only after a substantial delay, and is then rapidly upregulated to a high level to completely repress TFAP2C. We implemented this behavior by making $P$ be activated at a low level by $A$, and at a much higher level by autoactivation by making the production terms for $P$ additive with $\beta_{PP} > \beta_{NP}$. This separates production of $P$ into an initial slow regime of Activin-driven production before reaching a threshold level for autoactivation and then accumulating much faster, as described in the GRN construction section. In the initial regime, assuming no inhibition from $T$, we have

$$\frac{dP}{dt} = \frac{\beta_{NP} N^n}{K_{NP}^n + N^n} - \alpha P, \tag{8}$$

so that if $N > K_{NP}$, $P$ accumulates towards a steady-state level of $\mathrm{ss}_{low} = \beta_{NP}/\alpha$ before reaching its autoactivation threshold. Once the threshold has been reached, if Activin is removed, $P$ accumulates according to

$$\frac{dP}{dt} = \frac{\beta_{PP} P^n}{K_{PP}^n + P^n} - \alpha P. \tag{9}$$

We find the autoactivation-driven steady state by setting $dP/dt = 0$ in (9). Doing this and rearranging, we get

$$P^{n+1} - \frac{\beta_{PP}}{\alpha} P^n + K_{PP}^n P = 0.$$

Letting $n = 2$, we have

$$P\left(P^2 - \frac{\beta_{PP}}{\alpha} P - K_{PP}^2\right) = 0,$$

which has the solutions $P = 0$ and

$$P = \frac{\beta_{PP}/\alpha \pm \sqrt{(\beta_{PP}/\alpha)^2 - 4 K_{PP}^2}}{2},$$

so that

$$P = 0, \quad P = \tfrac{1}{2}\beta_{PP}/\alpha + \tfrac{1}{2}\sqrt{(\beta_{PP}/\alpha)^2 - 4 K_{PP}^2}$$

are stable fixed points and

$$P = \tfrac{1}{2}\beta_{PP}/\alpha - \tfrac{1}{2}\sqrt{(\beta_{PP}/\alpha)^2 - 4 K_{PP}^2}$$

is an unstable fixed point between them. Then, following the above analysis, the smaller non-zero root is the threshold, $\tau_P$, for autoactivation of P, and the larger root is the autoactivation-driven steady state, $\mathrm{ss}_{high}$. Setting $\tau_P$ and $\mathrm{ss}_{high}$ to desired values, we can write $K_{PP} = \sqrt{\mathrm{ss}_{high}\tau_P}$ and $\beta_{PP} = \alpha(\mathrm{ss}_{high} + \tau_P)$ in (3). In the case that the autoactivation threshold has been met and signaling remains on, the steady-state level reached will be $\mathrm{ss}_{high} + \mathrm{ss}_{low}$, but for $\mathrm{ss}_{high} \gg \mathrm{ss}_{low}$, we can take this to be approximately $\mathrm{ss}_{high}$. Because we also expected $A$ to initially accumulate slowly before being robustly autoactivated, we wrote the activation function of $A$ with the same structure as $P$, so that the analysis for autoactivation of $A$ is identical to that for $P$.

## Parameter considerations

For simplicity, and in the absence of concrete data on protein production and degradation rates, we set $\alpha_T = \alpha_S = \alpha_P = \alpha_A = \alpha$, taking each protein to degrade and dilute at approximately the same rate (and in fact, if we take the proteins to be stable the term $\alpha$ in each equation only describes dilution due to cell growth and division, which is constant across all proteins). Because the units of protein concentration in the simulation are arbitrary, we set $\beta_T = \beta_S = \alpha$ and $\mathrm{ss}_{high} = 1$ for both $P$ and $A$, so all concentrations vary between 0 and 1. We can additionally express the $\beta$ and $K$ parameters for $P$ and $A$ in terms of thresholds and steady states as

$$\beta_{NP} = \alpha \cdot ss_{lowP}$$

$$\beta_{BA} = \alpha \cdot ss_{lowA}$$

$$K_{PP} = \sqrt{ss_{high} \cdot \tau_P} = \sqrt{\tau_P}$$

$$K_{AA} = \sqrt{ss_{high} \cdot \tau_A} = \sqrt{\tau_A}$$

$$\beta_{PP} = \alpha(ss_{high} + \tau_P) = \alpha(1 + \tau_P)$$

$$\beta_{AA} = \alpha(ss_{high} + \tau_A) = \alpha(1 + \tau_A)$$

Because system behavior is similar for a reasonable range of $n$ from 2 to 4, we let it be the same for each equation and set $n = 2$. Then, we can rewrite (1)–(4) as

$$\frac{1}{\alpha}\frac{dT}{dt} = \frac{\left(B/K_{BT}\right)^2 + \left([TS]/K_{[TS]}\right)^2}{1 + \left(B/K_{BT}\right)^2 + \left([TS]/K_{[TS]}\right)^2} \cdot \frac{K_{PT}^2}{K_{PT}^2 + P^2} - T \tag{10}$$

$$\frac{1}{\alpha}\frac{dS}{dt} = \frac{\left(NB/K_{NBS}\right)^2 + \left([TS]/K_{[TS]}\right)^2}{1 + \left(NB/K_{NBS}\right)^2 + \left([TS]/K_{[TS]}\right)^2} \cdot \frac{K_{AS}^n}{K_{AS}^2 + A^n} - S \tag{11}$$

$$\frac{1}{\alpha}\frac{dP}{dt} = \left(\frac{ss_{lowP}N^2}{K_{NP}^2 + N^2} + \frac{(1 + \tau_P)P^2}{\tau_P + P^2}\right) \cdot \frac{K_{TP}^2}{K_{TP}^2 + T^2} - P \tag{12}$$

$$\frac{1}{\alpha}\frac{dA}{dt} = \left(\frac{ss_{lowA}B^2}{K_{BA}^2 + B^2} + \frac{(1 + \tau_A)A^2}{\tau_A + A^2}\right) \cdot \frac{K_{SA}^2}{K_{SA}^2 + S^2} - A \tag{13}$$

We also take BMP and Activin/NODAL activity to vary between minimum and maximum values of 0 and 1, and set activation thresholds in this range. Take $K_d$ to be large (initially $K_d = 100$), so that most available $X$ and $Y$ dimerize to form $[XY]$; note that the autoactivation threshold $K_{[XY]}$ can account for different values of $K_d$. The expected qualitative behavior of the model now mainly depends on thresholds for autoactivation and the thresholds for inhibition between $T$ and $P$ and between $S$ and $A$.

Finally, note that in our simulations we take BMP and NODAL-driven gene activation to rely on (sigmoidal) Hill functions where AM has a higher BMP threshold for activation than TFAP2C and PS has a higher NODAL threshold for activation than SOX17. This was a natural way to explain the observation that TFAP2C+ cells generally expand farther into the colony than the AmLC fate ring, and that SOX17 is expressed with very low Activin rescue doses while TFAP2C is only repressed by the proposed PS gene for the highest Activin dose. However, this is not an essential feature of the model, and similar qualitative behavior can be obtained if BMP and NODAL activate gene expression with first-order (non-sigmoidal) Hill functions depending on the relative inhibition strengths between TFAP2C and PS and between SOX17 and AM, as well as each gene's threshold for autoactivation.

Referring to the analysis of how a delay is imposed on robust activation of PS, recall that, neglecting inhibition, NODAL/Activin-mediated accumulation of $P$ follows (8), so that for a given signaling input level $N$, it approaches the steady state $ss_{low} = \frac{\beta_{NP}}{\alpha} \cdot \frac{N^n}{K_{NP}^n + N^n}$.

If activation by signaling is switch-like (high $n$), this can take on values of 0 or $\beta_{NP}/\alpha$, and whether the autoactivation threshold can be reached depends on whether the signaling input is above the threshold $K_{NP}$. If signaling-driven production is taken to be more graded (low $n$), then the steady state can take on a range of values between 0 and $\beta_{NP}/\alpha$. In this case, there is still a specific level of NODAL/Activin above which it is possible to reach the autoactivation threshold, but how long it takes to reach the threshold depends on the level of $N$. The signaling-mediated steady state is further lowered by the presence of inhibitors, which explains, for instance, why high BMP for the first 24 hr in the absence of NODAL/Activin is taken to sufficiently induce production of AM to a high level, but if BMP and Activin are concurrently applied, SOX17 represses production of AM, except at the colony edge at the lowest rescue dose. Further experimental work and analysis of the robustness of the system to perturbations in parameter values will shed light on which of these scenarios is more plausible.

## Parameter fitting

We determined gene expression patterns to which to fit the model by quantifying radial profiles of TFAP2C and SOX17 IF averaged over at least four micropatterned colonies for each of the 10 experimental conditions:

- WT cells treated with 50 ng/ml BMP4 (*Figure 1A*)
- WT cells treated with 50 ng/ml BMP4 for 20 hr and then switched to a high dose of BMPRi (*Figure 3—figure supplement 1C*)
- NODALKO cells treated with 50 ng/ml BMP4 and 0, 1, 3, 10, 30, or 100 ng/ml Activin (*Figure 4M*)
- NODALKO cells treated with 50 ng/ml BMP4 with 100 ng/ml Activin added only during the first 24 hr of differentiation or only after 24 hr (*Figure 4—figure supplement 2D*)

In each treatment condition, differentiation is for 42 hr after initial treatment.

To simulate the spatial patterning of hPSC colonies in each of these conditions, we considered 700 μm diameter colonies with signaling taken to be radially symmetric, with input BMP and NODAL/Activin signaling profiles as in *Figure 6A*. Keeping the values of $n$, $\alpha$, $\beta_T$, $\beta_S$, $ss_{high}$, and $K_d$ fixed, we optimized the remainder of the parameters with simulated annealing, as described in *Kirkpatrick et al., 1983* and *Bertsimas and Tsitsiklis, 1993*. The algorithm is as follows:

1. Randomly initialize a vector of parameters $\theta$
2. For a set number of iterations:
    a. Evaluate the error $E$ in model output across test conditions
    b. Perturb parameters: $\theta_{new} = \theta + \Delta\theta$, where $\Delta\theta \sim N(0, \Sigma)$
    c. Evaluate the error $E_{new}$ using parameters $\theta_{new}$, and set $\Delta E = E_{new} - E$
    d. If $\Delta E < 0$, set $\theta = \theta_{new}$. Otherwise, set $\theta = \theta_{new}$ with probability $\exp \frac{-\Delta E}{k_B T}$

This procedure simulates the reduction in energy of a system of atoms moving towards thermodynamic equilibrium at temperature $T$, with the possibility of accepting moves to regions of higher energy (error) preventing the algorithm from becoming trapped at a shallow local minimum. In analogy with physical annealing, the effective temperature is gradually reduced to zero so that that the acceptance criterion for steps to higher error becomes stricter as the algorithm progresses. To generate the results shown in *Figure 6* and *Figure 6—figure supplement 1*, we used the parameter values in *Appendix 1—table 1*.

**Appendix 1—table 1.** Model parameters.

| Parameter | Value | Meaning |
|---|---|---|
| n | 2 | Hill function coefficient |
| $\alpha$ | 0.1733 | Protein dilution + degradation rate |
| $(\beta_T, \beta_S)$ | 0.173 | Production rate for $T$ and $S$ |
| $ss_{high}$ | 1 | Autoactivation steady state for $P$ and $A$ |
| $(K_{BT}, K_{NBS}, K_{NP}, K_{BA})$ | (0.313, 0.338, 0.371, 1.2) | Signaling thresholds |
| $(K_{PT}, K_{AS}, K_{TP}, K_{SA})$ | (0.422, 0.373, 1.13, 1.14) | Inhibition thresholds |
| $(ss_{lowP}, ss_{lowA})$ | (0.269, 0.135) | Maximum signaling-driven protein level |
| $(\tau_P, \tau_A)$ | (0.376, 0.127) | Autoactivation thresholds |
| $K_{[TS]}$ | 0.5 | activation threshold for $[TS]$ on $T$ and $S$ |
| $K_d$ | 100 | dimerization constant for $[TS]$ |

