## [Editor Report]

This manuscript describes a powerful tissue culture model to study the early embryonic development of human primordial germ cells (PGC), the precursors of eggs and sperm. The study dissects the signaling pathways that direct the formation of PGC and provides important clues as to the origins of these cells during development.

---

## [Decision Letter]

**Decision letter after peer review:**

Thank you for submitting your article "Efficient differentiation of human primordial germ cells through geometric control reveals a key role for NODAL signaling" for consideration by *eLife*. Your article has been reviewed by 2 peer reviewers, and the evaluation has been overseen by a Reviewing Editor and Marianne Bronner as the Senior Editor. The reviewers have opted to remain anonymous.

Essential revisions:

The most important points below relate to further characterization of the hPGCLC cells. This should be as convincing as possible, to maximize the impact of your study.

1. In line 192, which is based on the data in Figure 1K, the authors stated that NANOG expression in hPGCLCs is lower than that in undifferentiated hESCs. It also looks like that POU5F1 expression in hPGCLCs is lower than that in undifferentiated hESCs. This somewhat contradicts with the observation made by preceding reports using a directed differentiation system such as those by Kojima et al., 2017, which showed POU5F1 and NANOG are substantially up-regulated during hPGCLC induction. Also, although the authors claimed that NANOG protein levels are higher in PGCLCs (Figure 1B and line 193), many cells residing in the PRDM1-positive area appeared to show relatively low NANOG. Are the SOX17/TFAP2C-positive cells induced in the micropatterned differentiation system somewhat different from more validated hPGCLCs in other systems?

2. Related to this point, Supplemental Figure 2F showed that with regard to the expression of selected highly variable genes, hPGCLCs in the micropatterned system exhibited low correlation to hPGCs in vivo, and showed even higher correlation to Non-Neural ectoderm (Amnion?), YS-mesoderm, Hypoblast, YS Endoderm, Advanced Mesoderm. Why is this?

3. The authors should demonstrate that "hPGCLCs" appeared in the micropatterned system truly represent hPGCLCs in a more rigorous manner; for example, by performing a direct transcriptome comparison between "hPGCLCs" in the micropatterned system and hPGCLCs induced in a directed differentiation system. This is critical, because if the "hPGCLCs" appeared in the micropatterned system are somewhat different from hPGCs in vivo, then, this system cannot be used for analyzing the hPGC specification mechanism.

4. In Figure 1D-G, which cells correspond to hPGCLCs? The relevant quadrants (top right quadrants) include cells expressing indicated proteins at diverse levels and do not appear to represent a uniform cell population.

5. Line 248: The authors use MAGIC to de-noise the scRNA-seq data. On the other hand, MAGIC has been known to often over-impute expression levels of key differentially expressed genes The authors need to make a careful use of the imputation method and provide an appropriate statement regarding this point.

6. Figure 2F and the paragraph starting from line 326: All the genes appeared to show continuous up-regulation during hPGCLC specification up to ~96 hour. The authors should show qPCR data of each gene in comparison to the expression levels of representative housekeeping genes in the same differentiation time point, but not to each gene level in undifferentiated ESCs, so that the readers can compare the absolute level of each gene in a more quantitative manner.

7. Line 680: Methods for cell culture and differentiation should be described in a more detailed manner so that the readers can at least understand the condition without referring to other papers. For example, inclusion of TGFβ in the culture media was stated in the main text, but not in the Methods section.

8. All the claims regarding the mechanism and manner of key cell-fate specifications were based on the authors' in vitro experimental system. It would be better to make discussion taking in vivo developmental context in humans or primates into account.

9. The authors should use colors that match the immunostaining colors for TFAP2C, SOX17 and FOXA2 in panels B, E, G and K of Figure 4 (as they do in other figures).

10. The images for TFAP2C and PRDM1 in Figure 5G are identical. It appears to be a mistake when assembling the figure as the merge image contains different blue and red channels.

11. It is not clear what the x-axis label "F" in Figure 5I refers to.

*Reviewer #1 (Public Review):*

Jo et al. use a combination of micropatterned differentiation, single cell RNA sequencing and pharmacological treatments to study primordial germ cell (PGC) differentiation starting from human pluripotent stem cells. Geometrical confinement in conjunction with a pre-differentiation step allowed the authors to reach remarkable differentiation efficiencies. While Minn et al. already reported the presence of PGC-like cells in micropatterned differentiating human cultures by scRNA-Seq (as acknowledged by the authors), the careful characterization of the PGC-like population using immunostainings and scRNA-Seq is a strength of the manuscript. The attempt at mechanistically dissecting the signaling pathways required for PGC fate specification is somehow weaker. The authors do not present sufficient evidence supporting the ability to specify PGC fate in the absence of Wnt signaling and the importance of the relative signaling levels of BMP to Nodal pathways; the wording of the text should be amended to better reflect the presented evidence or the authors should perform additional experiments to support these claims. The molecular characterization of why colonies confined to small areas differentiate much better would greatly increase the biological significance of the manuscript (the technical achievement of reaching such efficiency is impressive on its own).

The authors propose a mathematical model based on BMP and Nodal signaling that qualitatively recapitulate their experimental data. While the authors should be commended for providing examples of other simple models that do not fully recapitulate their data, it would have been nice to see an attempt at challenging quantitatively the model. In particular, the authors do not take advantage of the ability to explore in a more systematic manner the BMP/Nodal phase space with their system.

The authors' claim that PGCLC formation can be rescued by exogenous Activin when blocking endogenous Wnt production is surprising given the literature. The authors only show that they can restore a TFAP2C+SOX17+ population but do not actually stain for an established germ cell marker. It appears essential to perform a PRDM1 staining in these conditions (Figure 4A) to unambiguously identify this population.

The authors only provide weak evidence that the fates depend on the relative signaling levels of BMP and Nodal. Indeed, fewer cells acquire a fate the lower BMP concentration they use, including the fates marked by Sox17 expression. It would more convincing to show the assay of Figure 4F for a range of BMP concentrations at which the overall differentiation works sufficiently well.

*Reviewer #1 (Recommendations for the authors):*

The authors should use colors that match the immunostaining colors for TFAP2C, SOX17 and FOXA2 in panels B, E, G and K of Figure 4 (as they do in other figures).

The images for TFAP2C and PRDM1 in Figure 5G are identical. It appears to be a mistake when assembling the figure as the merge image contains different blue and red channels.

It is not clear what the x-axis label "F" in Figure 5I refers to.

*Reviewer #2 (Public Review):*

This manuscript uses a micropatterned differentiation system to explore the mechanism of human primordial germ cell-like cell (hPGCLC) specification and proposes a previously un-recognized role of NODAL signaling operating downstream of BMP signaling. The strength of the manuscript is the development of a simple in vitro system that is potentially suitable for exploring the mechanism of human primordial germ cell-like cell (hPGCLC) specification. The weakness of the manuscript is the lack of rigorous validation of the identity of the hPGCLCs appeared in the micropatterned differentiation system and the lack of the discussion relevant to the in vivo developmental mechanisms.

*Reviewer #2 (Recommendations for the authors):*

The manuscript by Jo et al. explores a signaling mechanism for human primordial germ cell-like cell (hPGCLCs) specification from human embryonic stem cells (hESCs) using a micropatterned differentiation system. This system allows relatively easy manipulations of culture conditions and is suitable for analyzing signaling and transcriptional requirements in relevant cell fate specifications. Furthermore, the authors developed a three dimensional (3D) image analysis pipeline to quantify the immunofluorescence analysis data for cell fate specification and based on such analyses under various experimental conditions, interrogated the differentiation process of hPGCLCs. Accordingly, the authors identified a previously unrecognized role of NODAL signaling that acts downstream of BMP signaling and replaces a requirement of WNT signaling for EOMES expression for hPGCLC specification. Based on these results, the authors formulated a simple mathematical model that takes only BMP and NODAL signaling into account and succeeds in recapitulating the authors' experimental outcome, and developed a simple and highly efficient (up to ~70%) hPGCLC induction system with a small diameter of cell micropatterning. Overall, this is a potentially interesting manuscript that may provide a robust system for the mechanistic understanding of the specification of hPGCLCs and other relevant cell types.

Following issues should be addressed to strengthen the claims by the authors and to improve the manuscript:

1. In line 192, which is based on the data in Figure 1K, the authors stated that NANOG expression in hPGCLCs is lower than that in undifferentiated hESCs. It also looks like that POU5F1 expression in hPGCLCs is lower than that in undifferentiated hESCs. This somewhat contradicts with the observation made by preceding reports using a directed differentiation system such as those by Kojima et al., 2017, which showed POU5F1 and NANOG are substantially up-regulated during hPGCLC induction. Also, although the authors claimed that NANOG protein levels are higher in PGCLCs (Figure 1B and line 193), many cells residing in the PRDM1-positive area appeared to show relatively low NANOG. Are the SOX17/TFAP2C-positive cells induced in the micropatterned differentiation system somewhat different from more validated hPGCLCs in other systems?

Related to this point, Supplemental Figure 2F showed that with regard to the expression of selected highly variable genes, hPGCLCs in the micropatterned system exhibited low correlation to hPGCs in vivo, and showed even higher correlation to Non-Neural ectoderm (Amnion?), YS-mesoderm, Hypoblast, YS Endoderm, Advanced Mesoderm. Why is this?

The authors should demonstrate that "hPGCLCs" appeared in the micropatterned system truly represent hPGCLCs in a more rigorous manner; for example, by performing a direct transcriptome comparison between "hPGCLCs" in the micropatterned system and hPGCLCs induced in a directed differentiation system. This is critical, because if the "hPGCLCs" appeared in the micropatterned system are somewhat different from hPGCs in vivo, then, this system cannot be used for analyzing the hPGC specification mechanism.

2. In Figure 1D-G, which cells correspond to hPGCLCs? The relevant quadrants (top right quadrants) include cells expressing indicated proteins at diverse levels and do not appear to represent a uniform cell population.

3. Line 248: The authors use MAGIC to de-noise the scRNA-seq data. On the other hand, MAGIC has been known to often over-impute expression levels of key differentially expressed genes (e.g., see Andrews, T. S. & Hemberg, M. False signals induced by single-cell imputation. F1000Res 7, 1740, doi:10.12688/f1000research.16613.2 (2018); Hou, W., Ji, Z., Ji, H. & Hicks, S. C. A systematic evaluation of single-cell RNA-sequencing imputation methods. Genome Biology 21, doi:10.1186/s13059-020-02132-x (2020)). The authors need to make a careful use of the imputation method and provide an appropriate statement regarding this point.

4. Figure 2F and the paragraph starting from line 326: All the genes appeared to show continuous up-regulation during hPGCLC specification up to ~96 hour. The authors should show qPCR data of each gene in comparison to the expression levels of representative housekeeping genes in the same differentiation time point, but not to each gene level in undifferentiated ESCs, so that the readers can compare the absolute level of each gene in a more quantitative manner.

5. Line 350: The authors used IWP2 as a WNT inhibitor. It would be better to use at least one more different inhibitor, since the chemical inhibitors are often not pathway-specific and show various off-target effects.

6. Line 680: Methods for cell culture and differentiation should be described in a more detailed manner so that the readers can at least understand the condition without referring to other papers. For example, inclusion of TGFβ in the culture media was stated in the main text, but not in the Methods section.

7. The terms such as "strikingly" and "surprisingly" appear too many times in the manuscript.

8. Line 206: Would it be appropriate to comment on the finding reported in the preprint as "established"?

9. All the claims regarding the mechanism and manner of key cell-fate specifications were based on the authors' in vitro experimental system. It would be better to make discussion taking in vivo developmental context in humans or primates into account.

---

## [Author Response]

Reviewer #1 (Public Review):Jo et al. use a combination of micropatterned differentiation, single cell RNA sequencing and pharmacological treatments to study primordial germ cell (PGC) differentiation starting from human pluripotent stem cells. Geometrical confinement in conjunction with a pre-differentiation step allowed the authors to reach remarkable differentiation efficiencies. While Minn et al. already reported the presence of PGC-like cells in micropatterned differentiating human cultures by scRNA-Seq (as acknowledged by the authors), the careful characterization of the PGC-like population using immunostainings and scRNA-Seq is a strength of the manuscript. The attempt at mechanistically dissecting the signaling pathways required for PGC fate specification is somehow weaker. The authors do not present sufficient evidence supporting the ability to specify PGC fate in the absence of Wnt signaling and the importance of the relative signaling levels of BMP to Nodal pathways; the wording of the text should be amended to better reflect the presented evidence or the authors should perform additional experiments to support these claims.

We thank the reviewer for this comment. As described in more detail in the responses below, we have significantly strengthened the evidence for the rescue of Wnt inhibition by exogenous Activin treatment and have nuanced our interpretation. We believe that our data suggest low levels of Wnt may be required directly for PGC competence, while much higher levels are required indirectly to induce Nodal, with Nodal signaling being the limiting factor for PGC specification under the reference condition with BMP4 treatment only. We describe this in detail in the manuscript but summarize it in Author response image 1 in a simplified diagram:

**Author response image 1. sa2fig1:** 

We have also carried out additional experiments that match model predictions demonstrating the importance of relative BMP and Nodal signaling levels and amended the text to reflect the evidence as suggested. More details are provided below.

The molecular characterization of why colonies confined to small areas differentiate much better would greatly increase the biological significance of the manuscript (the technical achievement of reaching such efficiency is impressive on its own).

We believe the mechanism by which cells confined to small colonies differentiate to PGCLCs more efficiently is explained by a larger fraction of the cells being exposed to the necessary levels of BMP and Nodal signaling. In large colonies BMP signaling was shown to be restricted to a distance of 50-100 μm from the colony edge through receptor localization and secretion of inhibitors (Etoc et al., Dev Cell 2016). From this one would expect that BMP signaling extends a similar distance from the edge in small colonies, so that a larger fraction of cells are receiving the BMP signal needed to differentiate to PGCLCs. Because it was not previously shown that the length scale of BMP signaling and downstream signals are preserved as colony size is reduced, we have now included an analysis of BMP signaling (pSmad1 levels) and Nodal signaling (nuclear Smad2/3 levels) as a function of colony size (Figure 5i-k). This confirms our hypothesis and provides a potential mechanism.

The authors propose a mathematical model based on BMP and Nodal signaling that qualitatively recapitulate their experimental data. While the authors should be commended for providing examples of other simple models that do not fully recapitulate their data, it would have been nice to see an attempt at challenging quantitatively the model. In particular, the authors do not take advantage of the ability to explore in a more systematic manner the BMP/Nodal phase space with their system.

We thank the reviewer for this suggestion. Experimentally we have now tested the effect of 5x5 = 25 different combinations of BMP and Activin doses on PGCLC differentiation. We then challenged the mathematical model to predict the ‘phase diagram’ corresponding to this data with good agreement (Figure 6f). It is important to note here that the model was fit using only data with 50ng/ml of BMP, making this a true prediction. We also point out that the phase diagram predicted in this way is different from the one shown in Figure 6d, not only because of the lower resolution, but because Figure 6f shows the steady state after uniform stimulation in space and time (i.e. the response on the very edge), whereas the predicted phase diagram shows average expression at 42h in a 100um range from the colony edge using the previously measured spatiotemporal gradients of BMP and Activin response. Finally, the data in Figure 6f shows mean expression levels as opposed to the percentage double positive cells for the same data in Figure 4q because our model does not simulate individual cells and noise, only allowing us to compare mean expression. We explain all this in the text now. As a minor change to facilitate comparison of data and model we have now plotted the concentrations of BMP and Activin in Figure 6 rather than the scaled model parameters from 0 to 1, we also further optimized the model parameters without qualitative changes.

The authors' claim that PGCLC formation can be rescued by exogenous Activin when blocking endogenous Wnt production is surprising given the literature. The authors only show that they can restore a TFAP2C+SOX17+ population but do not actually stain for an established germ cell marker. It appears essential to perform a PRDM1 staining in these conditions (Figure 4A) to unambiguously identify this population.

We have significantly extended our analysis of the effect of WNT inhibition and subsequent rescue of PGCs by Activin treatment. This includes staining for TFAP2C,NANOG,PRDM1 and staining for LEF1 as a measure of WNT signaling. Figure 4 and Figure 4—figure supplement 1 now also include treatment with IWR-1, a different small molecule inhibitor of WNT signaling, as well inhibition by IWR-1 and IWP2 at different times and different doses.

The authors only provide weak evidence that the fates depend on the relative signaling levels of BMP and Nodal. Indeed, fewer cells acquire a fate the lower BMP concentration they use, including the fates marked by Sox17 expression. It would more convincing to show the assay of Figure 4F for a range of BMP concentrations at which the overall differentiation works sufficiently well.

As suggested, we have now included a range of BMP concentrations. The reduction in PGCs at lower BMP doses is in line with our model and does not contradict a dependence on the relative signaling levels of BMP and Nodal by which we mean that optimal dose of Activin for PGCLC specification depends on the level of BMP and vice versa. We have amended the text to state this more clearly.

Reviewer #1 (Recommendations for the authors):The authors should use colors that match the immunostaining colors for TFAP2C, SOX17 and FOXA2 in panels B, E, G and K of Figure 4 (as they do in other figures).

We thank the reviewer for pointing this out, the colors match now.

The images for TFAP2C and PRDM1 in Figure 5G are identical. It appears to be a mistake when assembling the figure as the merge image contains different blue and red channels.

We thank the reviewer for pointing out this mistake and we have corrected it.

It is not clear what the x-axis label "F" in Figure 5I refers to.

The revised manuscript no longer contains this panel but shows scatterplots for two conditions instead (Figure 5fg).

Reviewer #2 (Public Review):This manuscript uses a micropatterned differentiation system to explore the mechanism of human primordial germ cell-like cell (hPGCLC) specification and proposes a previously un-recognized role of NODAL signaling operating downstream of BMP signaling. The strength of the manuscript is the development of a simple in vitro system that is potentially suitable for exploring the mechanism of human primordial germ cell-like cell (hPGCLC) specification. The weakness of the manuscript is the lack of rigorous validation of the identity of the hPGCLCs appeared in the micropatterned differentiation system and the lack of the discussion relevant to the in vivo developmental mechanisms.Reviewer #2 (Recommendations for the authors):The manuscript by Jo et al. explores a signaling mechanism for human primordial germ cell-like cell (hPGCLCs) specification from human embryonic stem cells (hESCs) using a micropatterned differentiation system. This system allows relatively easy manipulations of culture conditions and is suitable for analyzing signaling and transcriptional requirements in relevant cell fate specifications. Furthermore, the authors developed a three dimensional (3D) image analysis pipeline to quantify the immunofluorescence analysis data for cell fate specification and based on such analyses under various experimental conditions, interrogated the differentiation process of hPGCLCs. Accordingly, the authors identified a previously unrecognized role of NODAL signaling that acts downstream of BMP signaling and replaces a requirement of WNT signaling for EOMES expression for hPGCLC specification. Based on these results, the authors formulated a simple mathematical model that takes only BMP and NODAL signaling into account and succeeds in recapitulating the authors' experimental outcome, and developed a simple and highly efficient (up to ~70%) hPGCLC induction system with a small diameter of cell micropatterning. Overall, this is a potentially interesting manuscript that may provide a robust system for the mechanistic understanding of the specification of hPGCLCs and other relevant cell types.Following issues should be addressed to strengthen the claims by the authors and to improve the manuscript:1. In line 192, which is based on the data in Figure 1K, the authors stated that NANOG expression in hPGCLCs is lower than that in undifferentiated hESCs. It also looks like that POU5F1 expression in hPGCLCs is lower than that in undifferentiated hESCs. This somewhat contradicts with the observation made by preceding reports using a directed differentiation system such as those by Kojima et al., 2017, which showed POU5F1 and NANOG are substantially up-regulated during hPGCLC induction. Also, although the authors claimed that NANOG protein levels are higher in PGCLCs (Figure 1B and line 193), many cells residing in the PRDM1-positive area appeared to show relatively low NANOG. Are the SOX17/TFAP2C-positive cells induced in the micropatterned differentiation system somewhat different from more validated hPGCLCs in other systems?

We believe that the discrepancies are due to timing. PGCLCs in our manuscript at 42h are likely less mature than those at day 2 (48h) in Kojima et al. We examined PGCLCs in our system a day later and found that they express NANOG and POU5F1 at much higher levels than pluripotent cells, similar to those in Kojima at 48h. We now show this with immunofluorescence staining for NANOG and POU5F1 at 48h, 72h, 96h in Figure 2hg and Figure 2—figure supplement 2, and have added a discussion regarding NANOG and POU5F1 levels to reflect these observations. We also collected scRNA-seq data at 48, 72, and 96 hours and similarly found that NANOG and POU5F1 substantially increase over time in PGCLCs relative to pluripotent cells, including from 42h to 48h (we show the data and provide further discussion in the next section of this response). The fact that our PGCLCs initially express NANOG and POU5F1 at levels similar to pluripotent epiblast cells is consistent with immunofluorescence data on nascent PGCs in cynomolgus monkeys, where the earliest PGCs appear to express NANOG at levels similar to the epiblast (Sasaki et al. Dev Cell 2016 Figure S5B).

The authors should demonstrate that "hPGCLCs" appeared in the micropatterned system truly represent hPGCLCs in a more rigorous manner; for example, by performing a direct transcriptome comparison between "hPGCLCs" in the micropatterned system and hPGCLCs induced in a directed differentiation system. This is critical, because if the "hPGCLCs" appeared in the micropatterned system are somewhat different from hPGCs in vivo, then, this system cannot be used for analyzing the hPGC specification mechanism.

In summary: We carried out scRNA-seq analysis comparing our micropatterned hPGCLCs to ‘conventional’ hPGCLCs (Chen et al., Cell Rep 2019), and to PGCs from the CS7 human gastrula (Tyser et al., Nature 2021). We conclude that our PGCLCs appear at least as similar to in vivo PGCs as other hPSC-derived PGCLCs. We discuss the details of our analysis below.

Extending our analysis of NANOG and POU5F1, we first compared scRNA-seq data that we collected for micropatterned hPGCLCs at 42, 48,72, and 96h across the full panel of marker genes used to establish PGC identity in Figure 4e of Kobayashi et al., Nature 2017. We also added PDPN and KIT to this panel as they are also commonly used cell surface markers for PGCs (e.g. Sasaki et al. Dev Cell 2016). In addition to NANOG and POU5F1 we also see a clear increase in other early PGC markers over time. We share this data in Author response image 2 but have not included it in the main manuscript because the full dataset is incredibly rich and describing it in detail is beyond the scope of this manuscript, so we are in the process of writing a separate manuscript about it.

**Author response image 2. sa2fig2:** Micropatterned PGCLCs.

We see a similar gene signature as in Kobayashi d4 hPGC+Cy with a few discrepancies. First PGCs in Kobayashi also appear to express some mesoderm and endoderm markers, which may represent imperfect FACS sorting since their data is bulk RNA-seq, or represent a difference in the induction strategy. Furthermore, we appear to lack DND1 and DPPA3 and SYCP3 in comparison to their d4 hPGC+Cy. This again may reflect a difference in experimental setup, for example we see small amounts of DDPA3 transcripts in the mesoderm but not the early PGCs and the bulk RNAseq may combine those. Therefore, we also compared our scRNA-seq data with that for hPGCLCs from Chen et al. Cell Rep 2019 and the Tyser et al. Nature 2021 CS7 human gastrula (in vivo data) and processed each sample in the same way (Author response image 3):

**Author response image 3. sa2fig3:** 

We see that the transcriptional profiles of the PGCLCs from Chen et al. are very similar to ours and to the profile from the CS7 human gastrula. It is difficult to say with certainty whether differences are due to technical limitations or biology. For example, we appear to be missing low levels of DND1 relative to Chen et al., but in Author response image 2 we are showing z-scores, i.e. relative expression compared to the other cell populations in the dataset. If we look at absolute expression in Author response image 4 we see that DND1 is barely expressed in Chen et al. and for us may have simply been below the detection threshold, since sequencing depth in scRNAseq is typically on the order of one read per gene per cell. As another example, one might conclude from Author response image 3 that our PGCLCs upregulate KLF4 and TFAP2C more robustly than Chen et al., but again that is not supported when looking at absolute expression. Overall, we prefer to not overinterpret the data and view the expression profiles as very similar.

**Author response image 4. sa2fig4:** 

In that regard it is also striking that the human embryo data lacks expression significant expression of PRDM1 in the PGCs. This may again simply be due to noise, especially since their PGC population is only 8 cells. It may also be a clustering error: PRDM1+TFAP2C+ cells are present in that dataset but not annotated as PGC but rather as non-neural/amniotic ectoderm. Again, similar considerations apply to other differences.

Related to this point, Supplemental Figure 2F showed that with regard to the expression of selected highly variable genes, hPGCLCs in the micropatterned system exhibited low correlation to hPGCs in vivo, and showed even higher correlation to Non-Neural ectoderm (Amnion?), YS-mesoderm, Hypoblast, YS Endoderm, Advanced Mesoderm. Why is this?

We thank the reviewer for pointing this out. We found that there was an error in the figure since the correlation was calculated on the wrong set of highly variable genes. It is now calculated on the intersection of the top 10% most highly variable genes from each data sets. With the new calculation PGCLCs in 42h micropatterned colonies correlate most strongly to PGCs, but the PGCLCs still show relatively low correlation with in vivo PGC and relatively high correlation with non-neural ectoderm (amniotic ectoderm) and endoderm. In summary: we believe that this is also due to a combination of (1) our PGCLCs at 42h being nascent / relatively immature and (2) because of technical limitations in how correlations between datasets are being calculated.

To support that low correlation at 42h has to do with the developmental stage of our PGCLCs relative to the CS7 PGCs, we calculated correlation in an identical way for our 48, 72, 96h timepoints (Author response image 5). We see dramatic increases in correlation between PGCLCs and PGCs and relative decreases in correlation with other cell types. The correlation of PGCLCs at 42h with non-neural/amniotic ectoderm and endoderm may hint at shared early transcriptional profiles, consistent with our model placing PGC in between these cell types (Figure 6d).

**Author response image 5. sa2fig5:** 

While relative correlations in these Author response images are suggestive, absolute correlation should not be quantitatively compared with numbers elsewhere, since correlations are calculated on normalized expression (z-score) and are highly sensitive to the overall cell content of the dataset (e.g. the PGCLC transcriptome after normalization will be different depending on whether endodermal cells are present or not in the same dataset) as well as various preprocessing and integration steps. Our previous calculations of correlation in the manuscript did not integrate datasets because we have found results to strongly depend on the integration method. However, to illustrate the effect we did integrate our 42h micropattern data with the human gastrula data using the most popular method: Seurat and have included this in Figure 1—figure supplement 2f. As might be expected correlations after integration are much higher, with PGCLC-PGC correlation going from 0.24 to 0.73.

To give basis of comparison of our PGCLCs to other PGCLCs, we also calculated the correlation matrices of the data from Chen et al. Cell Rep 2019 using the exact same methods as ours, without integration (Author response image 6). These should be compared to our correlation matrices in Author response image 5 and in Figure 2f, left. We did not fully annotate this dataset and only labeled the PGC cluster, which is unambiguous.

**Author response image 6. sa2fig6:** 

We see that by 96h, PGCLC-PGC correlations are similar to, but in fact somewhat lower than for our system. We conclude that our PGCLCs appear at least as similar to in vivo PGCs as other hPSC-derived PGCLCs.

2. In Figure 1D-G, which cells correspond to hPGCLCs? The relevant quadrants (top right quadrants) include cells expressing indicated proteins at diverse levels and do not appear to represent a uniform cell population.

Indeed, the expression levels are not uniform in each cell population. Gene expression levels are heterogeneous because of biological noise and on top of that there is measurement noise. Moreover, because of the spatiotemporal dynamics of signaling, cells may be at different stages of differentiation in different positions adding more heterogeneity. Nevertheless, coloring scatterplots for density shows clearly defined populations when population sizes are similar, as happens on the small micropatterns, and we now show this in Figure 5—figure supplement 1c (if another population is much larger as is the case for large micropatterns, it is hard to see the density in the smaller PGC cluster). We believe our observations are generally expected and have no knowledge of any study showing single cell quantitative analysis of protein expression in this manner where the studied cell populations are found to have more uniform expression.

3. Line 248: The authors use MAGIC to de-noise the scRNA-seq data. On the other hand, MAGIC has been known to often over-impute expression levels of key differentially expressed genes (e.g., see Andrews, T. S. & Hemberg, M. False signals induced by single-cell imputation. F1000Res 7, 1740, doi:10.12688/f1000research.16613.2 (2018); Hou, W., Ji, Z., Ji, H. & Hicks, S. C. A systematic evaluation of single-cell RNA-sequencing imputation methods. Genome Biology 21, doi:10.1186/s13059-020-02132-x (2020)). The authors need to make a careful use of the imputation method and provide an appropriate statement regarding this point.

We thank the reviewer for this point of caution. We are aware of the risks of denoising/imputation and believe we did make careful use of it. None of our conclusions rely on MAGIC. As stated explicitly in the methods section, MAGIC was used for two purposes only:

1) Visualization in Figure 1. It is easier to see the gene expression patterns on PHATE maps after some smoothing with MAGIC. We have now included the same data without MAGIC in Figure 1—figure supplement 2d and have also include raw data in Figure 1—figure supplement 2e.

2) Relationships between pairs of genes in Figure 1—figure supplement 1k,n. While none of our conclusions rest on these figures, we believe that they actually demonstrate the value of imputation, because while raw single cell RNA-seq data is so noisy one cannot discern any relationship between any pair of genes, denoising yields relationships between genes that closely resemble those found from immunofluorescence data providing cross-validation for both approaches. We also discuss the differences found in the main text and state imputation artefacts are one possible explanation for these differences. Again, we have now included raw data in Figure 1—figure supplement 1lo to emphasize this.

We only used magic for these two purposes and have revised the text to make this clear. For all other analysis, such as cluster comparison with heatmaps (e.g. Figure 1k), and differential expression analysis, we used the raw data, without MAGIC.

4. Figure 2F and the paragraph starting from line 326: All the genes appeared to show continuous up-regulation during hPGCLC specification up to ~96 hour. The authors should show qPCR data of each gene in comparison to the expression levels of representative housekeeping genes in the same differentiation time point, but not to each gene level in undifferentiated ESCs, so that the readers can compare the absolute level of each gene in a more quantitative manner.

We have now included CT values relative to GAPDH in Figure 2—figure supplement 1e.

5. Line 350: The authors used IWP2 as a WNT inhibitor. It would be better to use at least one more different inhibitor, since the chemical inhibitors are often not pathway-specific and show various off-target effects.

As also described above in response to a comment by reviewer 2, we have significantly extended and nuanced our analysis of the effect of WNT inhibition and subsequent rescue of PGCs by Activin treatment. Figure 4 and Figure 4—figure supplement 1 now include treatment with IWR-1, a different small molecule inhibitor of WNT signaling, as well inhibition by IWR-1 and IWP2 at different times and different doses.

6. Line 680: Methods for cell culture and differentiation should be described in a more detailed manner so that the readers can at least understand the condition without referring to other papers. For example, inclusion of TGFβ in the culture media was stated in the main text, but not in the Methods section.

We have now explicitly stated in the methods section that mTeSR1 contains FGF2 and TGF*β* and added additional description of the differentiation protocol.

7. The terms such as "strikingly" and "surprisingly" appear too many times in the manuscript.

We have removed several instances of these and similar words.

8. Line 206: Would it be appropriate to comment on the finding reported in the preprint as "established"?

We have changed “established” to “suggested”. The work we referred to is now published in Nature Communications.

9. All the claims regarding the mechanism and manner of key cell-fate specifications were based on the authors' in vitro experimental system. It would be better to make discussion taking in vivo developmental context in humans or primates into account.

We agree and thank the reviewer for this comment. We have added a paragraph relating our results to in vivo findings.

References

Chen, Di, Na Sun, Lei Hou, Rachel Kim, Jared Faith, Marianna Aslanyan, Yu Tao, et al. 2019. “Human Primordial Germ Cells Are Specified From Lineage-Primed Progenitors..” Cell Reports 29 (13): 4568–4582.e5. doi:10.1016/j.celrep.2019.11.083.

Etoc, Fred, Jakob Metzger, Albert Ruzo, Christoph Kirst, Anna Yoney, M Zeeshan Ozair, Ali H Brivanlou, and Eric D Siggia. 2016. “A Balance Between Secreted Inhibitors and Edge Sensing Controls Gastruloid Self-Organization..” Developmental Cell 39 (3): 302–15. doi:10.1016/j.devcel.2016.09.016.

Kobayashi, Toshihiro, Haixin Zhang, Walfred W C Tang, Naoko Irie, Sarah Withey, Doris Klisch, Anastasiya Sybirna, et al. 2017. “Principles of Early Human Development and Germ Cell Program From Conserved Model Systems..” Nature 546 (7658): 416–20. doi:10.1038/nature22812.

Kojima, Yoji, Kotaro Sasaki, Shihori Yokobayashi, Yoshitake Sakai, Tomonori Nakamura, Yukihiro Yabuta, Fumio Nakaki, et al. 2017. “Evolutionarily Distinctive Transcriptional and Signaling Programs Drive Human Germ Cell Lineage Specification From Pluripotent Stem Cells..” Cell Stem Cell 21 (4): 517–532.e5. doi:10.1016/j.stem.2017.09.005.

Sasaki, Kotaro, Tomonori Nakamura, Ikuhiro Okamoto, Yukihiro Yabuta, Chizuru Iwatani, Hideaki Tsuchiya, Yasunari Seita, et al. 2016. “The Germ Cell Fate of Cynomolgus Monkeys Is Specified in the Nascent Amnion..” Developmental Cell 39 (2): 169–85. doi:10.1016/j.devcel.2016.09.007.

Tyser, R.C.V., Mahammadov, E., Nakanoh, S. et al. Single-cell transcriptomic characterization of a gastrulating human embryo. Nature 600, 285–289 (2021). https://doi.org/10.1038/s41586-021-04158-y